# Enhancing Robust Fairness via Confusional Spectral Regularization

Gaojie Jin[1,2], Sihao Wu[3], Jiaxu Liu[3], Tianjin Huang[2], and Ronghui Mu[2]

[1]The Key Laboratory of System Software (Chinese Academy of Sciences) and State Key Laboratory of
Computer Science, Institute of Software, Chinese Academy of Sciences, Beijing, China
[2]Department of Computer Science, University of Exeter, Exeter, UK
[3]Department of Computer Science, University of Liverpool, Liverpool, UK
Corresponding to: gaojie.jin.kim@gmail.com

## Abstract

Recent research has highlighted a critical issue known as "robust fairness", where robust accuracy varies significantly across different classes, undermining the reliability of deep neural networks (DNNs). A common approach to address this has been to dynamically reweight classes during training, giving more weight to those with lower empirical robust performance. However, we find there is a divergence of class-wise robust performance between training set and testing set, which limits the effectiveness of these explicit reweighting methods, indicating the need for a principled alternative. In this work, we derive a robust generalization bound for the worst-class robust error within the PAC-Bayesian framework, accounting for unknown data distributions. Our analysis shows that the worst-class robust error is influenced by two main factors: the spectral norm of the empirical robust confusion matrix and the information embedded in the model and training set. While the latter has been extensively studied, we propose a novel regularization technique targeting the spectral norm of the robust confusion matrix to improve worst-class robust accuracy and enhance robust fairness. We validate our approach through comprehensive experiments on various datasets and models, demonstrating its effectiveness in enhancing robust fairness.

## 1 Introduction

Deep neural networks, spanning a diverse array of domains and applications, have shown impressive abilities to learn from training data and generalize effectively to new, unseen data. However, recent studies have uncovered a notable weakness in these DNNs – their vulnerability to subtle, often undetectable "adversarial attacks" (Biggio et al., 2013; Szegedy et al., 2013). It has been discovered that even slight perturbations to the input, typically imperceptible to humans, can drastically mislead the networks, resulting in significant prediction errors (Goodfellow et al., 2015; Wu et al., 2020a). Adversarial training is widely acknowledged as a potent defense against adversarial attacks (Athalye et al., 2018). Building on this, numerous studies have further developed and refined this approach to bolster robustness, contributing significant advancements in the field (Wu et al., 2020b; Lee et al., 2020; Cui et al., 2021; Jin et al., 2022a; Zhang et al., 2019; 2023).

In traditional machine learning, the definition of fairness (Agarwal et al., 2018; Hashimoto et al., 2018) may differ from the concept of robust fairness that we aim to address in this work, which focuses on mitigating fairness issues in the context of adversarial attacks (i.e., improving worst-class robust accuracy). The complex interplay between robustness and fairness, as highlighted by Xu et al. (2021), shows that the robust accuracy of a model can significantly differ across various categories or classes. For example, a traffic detection system that achieves impressive overall robust accuracy in detecting road objects. Despite its general success, the system could exhibit high robustness for certain categories like inanimate objects, yet be less robust when identifying critical categories, such as "humans". Such unevenness in robustness, where some categories are less protected, poses a risk

| Generalization Bound of Confusion Matrix over Gibbs Classifiers (Thm. 2.1) | → First Step | Generalization Bound over Deterministic Classifiers (Lems. 3.2 and 3.3) | → Second Step (I) | Robust Generalization Bound of Confusion Matrix (Lem. 3.4) | → Second Step (II) | Robust Generalization Bound for Worst-class Robust Error (Prop. 3.1) |
|---|---|---|---|---|---|---|

Figure 1: Illustration of the theoretical framework: worst-class robust generalization bound. Under this framework, a standard generalization bound over confusion matrix is extended to a robust generalization bound for the worst-class robust error.

to both drivers and pedestrians. Therefore, it is crucial to establish uniformly high and fair model performance against adversarial attacks.

Several works have proposed enhancing model robust fairness through explicit reweighting strategies during adversarial training, where classes are reweighted based on their robust performance over the training set (Xu et al., 2021; Li & Liu, 2023; Zhang et al., 2024). However, our analysis reveals that the robust class-wise performance over training set does not consistently align with that of testing set. As shown in Fig. 2 (left), the class exhibiting the worst robust performance on the training set may not be the same as the one on the testing set. Furthermore, Fig. 2 (right) demonstrates that these explicit reweighting approaches, e.g., FRL (Xu et al., 2021) and FAAL (Zhang et al., 2024), can actually exacerbate the training-test divergence, consequently limiting their ability to optimize worst-class robust accuracy. This misalignment between training and test robust performance across classes, combined with the absence of rigorous theoretical foundations, motivates our development of a principled alternative to improve worst-class robust performance. As shown in Fig. 2 (right), our developed method maintains significantly lower training-test divergence (higher training-test correlation) compared to explicit reweighting approaches. Note that our method is not primarily designed to address the training-test divergence, which may be inherent to the dataset, but rather to avoid explicit reweighting that can exacerbate this divergence and limit effectiveness.

In this work, we derive a robust generalization bound for the worst-class robust error within the PAC-Bayesian framework. The PAC-Bayesian approach, introduced by McAllester (1999), is designed to provide probably approximately correct (PAC) guarantees to "Bayesian-like" learning algorithms (e.g., a Gibbs classifier defined on a posterior distribution). As illustrated in Fig. 1, our first step is to transfer the bound in Morvant et al. (2012) for a Gibbs classifier to bound the spectral norm of the confusion matrix for a deterministic classifier. We analytically extend this bound by incorporating the structural information of a deterministic model. This process is similar to that of Neyshabur et al. (2017b), but with a key difference: we establish a chain derivation over confusion matrices to construct the bound. Secondly, leveraging the local perturbation bound

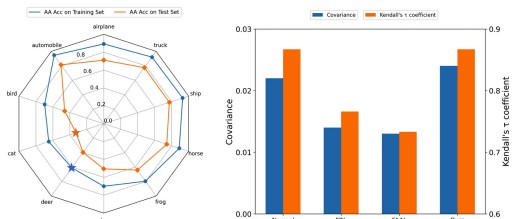

Figure 2: **Left**: Class-wise AA (Auto Attack) accuracy of the adversarially trained WideResNet-28-10 model on CIFAR-10. The star points are the worst-class AA accuracy on training set and testing set. **Right:** The covariance (blue) and Kendall rank correlation (orange) of class-wise AA accuracy between training set and testing set for normal adversarially trained WRN-28-10 model, reweighting methods (FRL and FAAL) fine-tuned models, and our method fine-tuned model.

approach from Xiao et al. (2023), we adapt the generalization bound over confusion matrices to the robust setting, accounting for adversarial perturbations. Through the relation of $\ell_1$ matrix norm and spectral norm, we finally provide a PAC-Bayesian bound for the worst-class robust error.

This bound unveils that the worst-class robust error is constrained by two key components: the spectral norm of the empirical robust confusion matrix and the information encapsulated within the model's architecture and training set. While the latter component has been extensively studied and leveraged to improve generalization or robust generalization (Yoshida & Miyato, 2017; Farnia et al., 2019), we propose a complementary approach to tackle this challenge. That is, we introduce a regularization technique that directly targets the spectral norm of the empirical robust confusion matrix, with the aim of improving worst-class robust accuracy and further enhancing robust fairness across different classes. To validate the effectiveness of our proposed method to improve worst-class performance, we conduct extensive experiments on various datasets. Our empirical evaluations demonstrate the superior performance of our approach in improving worst-class robust accuracy,

ensuring more equitable and reliable model predictions under adversarial conditions, even for the most vulnerable classes. To summarize, the contributions of this work are as follows:

- By extending the principled concepts of PAC-Bayesian generalization analysis to the domain of robust fairness, we develop a robust generalization framework that enables us to bound the worst-class robust error, as detailed in Sec. 3. To the best of our knowledge, this work represents the first endeavor to develop a PAC-Bayesian framework to characterize the worst-class robust error across different classes.

- As a by-product, leveraging the insights gleaned from our theoretical results, we propose an effective and principled method to enhance robust fairness by introducing a spectral regularization term on the confusion matrix. Empirically, extensive experiments on CIFAR-10/100 and Tiny-ImageNet datasets have been conducted to demonstrate the effectiveness of our method. (Sec. 4)

## 2 PRELIMINARIES

**Basic setting.** Consider an input set $\mathcal{X}_B$ defined as $\mathcal{X}_B = \{\mathbf{x} \in \mathbb{R}^d \mid \sum_{i=1}^d x_i^2 \leq B^2\}$ and a label set $\mathcal{Y} = \{1, \ldots, d_y\}$. Let $\mathcal{S} = \{(\mathbf{x}_1, y_1), \ldots, (\mathbf{x}_m, y_m)\}$ represent a training dataset comprising $m$ samples independently and identically drawn from an underlying, fixed but unknown distribution $\mathcal{D}$ on $\mathcal{X}_B \times \mathcal{Y}$. We define $f_{\mathbf{w}} : \mathcal{X}_B \to \mathcal{Y}$ as the learning function parameterized by weights $\mathbf{w}$, where $\mathbf{w}$ are real-valued weights for functions mapping $\mathcal{X}_B$ to $\mathcal{Y}$. We call the set of classifiers (or the set of classifier weights), i.e., $\mathcal{H}$, as the hypotheses. Let $\ell : \mathcal{H} \times \mathcal{X}_B \times \mathcal{Y} \to \mathbb{R}^+$ be the loss function used in the training.

The function $f_{\mathbf{w}}$ is structured as an $n$-layer neural network, each layer equipped with $h$ hidden units and utilizing the ReLU activation function $\phi(\cdot)$. The weight matrices for the entire model and $l$-th layer are denoted by $\mathbf{W}$ and $\mathbf{W}_l$ (capital letters), respectively, with $\mathbf{w}$ and $\mathbf{w}_l$ representing their vectorized forms (lowercase letters, i.e., $\mathbf{w} = \text{vec}(\mathbf{W})$). The operation of the function through the layers is formulated as $f_{\mathbf{w}}(\mathbf{x}) = \mathbf{W}_n \phi(\mathbf{W}_{n-1} \ldots \phi(\mathbf{W}_1 \mathbf{x}) \ldots)$, where $f_{\mathbf{w}}^{(1)}(\mathbf{x}) = \mathbf{W}_1 \mathbf{x}$ and subsequent layers are defined recursively as $f_{\mathbf{w}}^{(l)}(\mathbf{x}) = \mathbf{W}_l \phi(f_{\mathbf{w}}^{(l-1)}(\mathbf{x}))$. For simplicity, bias are integrated into the weight matrices. We denote $\|\mathbf{W}\|_2$ as the spectral norm of $\mathbf{W}$, represents the largest singular value. Additionally, $\|\mathbf{W}\|_F$ is the Frobenius norm of the weight matrix and $\|\mathbf{w}\|_p$ is the $\ell_p$ norm of the weight vector, respectively.

We define the empirical and the true confusion matrices of $f_{\mathbf{w}}$ by respectively $\mathcal{C}_{\mathcal{S}}^{f_{\mathbf{w}}} = (\hat{c}_{ij})_{1 \leq i,j \leq d_y}$ and $\mathcal{C}_{\mathcal{D}}^{f_{\mathbf{w}}} = (c_{ij})_{1 \leq i,j \leq d_y}$ such that

$$
\begin{aligned}
\forall(i,j), \hat{c}_{ij} &:= \begin{cases} 0 & i = j \\ \sum_{q=1}^m \frac{1}{m_j} \mathbb{1}(f_{\mathbf{w}}(\mathbf{x}_q)[i] \geq \max_{i \neq i'} f_{\mathbf{w}}(\mathbf{x}_q)[i']) \, \mathbb{1}(y_q = j) & \text{else,} \end{cases} \\
\forall(i,j), c_{ij} &:= \begin{cases} 0 & i = j \\ \mathbb{P}_{(\mathbf{x},y) \sim \mathcal{D}}(f_{\mathbf{w}}(\mathbf{x})[i] \geq \max_{i \neq i'} f_{\mathbf{w}}(\mathbf{x})[i'] \mid y = j) & \text{else,} \end{cases}
\end{aligned}
\tag{1}
$$

where $m_j$ is the number of samples with label $j$ in the training set $\mathcal{S}$. In previous works such as Neyshabur et al. (2017b); Farnia et al. (2019), the PAC-Bayesian generalization analysis for a DNN is conducted on the margin loss. Following the margin setting, we consider any positive margin $\gamma$ in this work and define the empirical margin confusion matrix $\mathcal{C}_{\mathcal{S},\gamma}^{f_{\mathbf{w}}} = (\hat{c}_{ij}^{\gamma})_{1 \leq i,j \leq d_y}$ as, $\forall(i,j)$,

$$
\hat{c}_{ij}^{\gamma} := \begin{cases} 0 & i = j \\ \sum_{q=1}^m \frac{1}{m_j} \mathbb{1}(f_{\mathbf{w}}(\mathbf{x}_q)[y_q] \leq \gamma + f_{\mathbf{w}}(\mathbf{x}_q)[i]) \mathbb{1}(y_q = j) \mathbb{1}(\arg\max_{i' \neq y_q} f_{\mathbf{w}}(\mathbf{x}_q)[i'] = i) & \text{else,} \end{cases}
$$

where $\mathbb{1}[a \leq b] = 1$ if $a \leq b$, else $\mathbb{1}[a \leq b] = 0$.

**Remark 1** *Following Neyshabur et al. (2017b), $\gamma$ is an auxiliary variable that aids in the development of the theory, setting $\gamma = 0$ corresponds to the normal empirical confusion matrix in (1). Note that $\gamma$ only exists in the confusion matrix over training data $\mathcal{S}$, ensuring that the classifier operates independently of the label over unseen data $\mathcal{D}$.*

**PAC-Bayes.** PAC-Bayes (McAllester, 1999; 2003) offers a tight upper bound on the generalization ability of a stochastic classifier called the Gibbs classifier (Laviolette & Marchand, 2005), which is defined on a posterior distribution $Q$ over $\mathcal{H}$. This framework also provides a generalization guarantee for the $Q$-weighted majority vote classifier, which is associated with this Gibbs classifier and assigns labels to input instances based on the most probable output from the Gibbs classifier (Lacasse et al., 2006; Germain et al., 2015; Jin et al., 2020; 2022b). The bound is primarily determined by the Kullback-Leibler divergence (KL) between the posterior distribution $Q$ and the prior distribution $P$ of weights (classifiers). Within this setting, we define the Gibbs classifiers over the posterior $Q$ of the type $f_{\mathbf{w}+\mathbf{u}}$ (Neyshabur et al., 2017b), where $\mathbf{u}$ is a random variable potentially influenced by the training data and $\mathbf{w}$ is the deterministic weights. In this case, we can define the true and the empirical (margin) confusion matrices of Gibbs classifier $f_{\mathbf{w}+\mathbf{u}}$ respectively by

$$\mathcal{C}_{\mathcal{D}}^Q = \mathbb{E}_{\mathbf{u}}\mathcal{C}_{\mathcal{D}}^{f_{\mathbf{w}+\mathbf{u}}}; \quad \mathcal{C}_{\mathcal{S}}^Q = \mathbb{E}_{\mathbf{u}}\mathcal{C}_{\mathcal{S}}^{f_{\mathbf{w}+\mathbf{u}}}; \quad \mathcal{C}_{\mathcal{S},\gamma}^Q = \mathbb{E}_{\mathbf{u}}\mathcal{C}_{\mathcal{S},\gamma}^{f_{\mathbf{w}+\mathbf{u}}}. \tag{2}$$

Morvant et al. (2012) proposes a PAC-Bayesian bound for the generalization risk of the Gibbs classifier using the confusion matrix of multi-class labels, a framework we also adopt in this work.

**Theorem 2.1 (Morvant et al. (2012))** *Consider a training dataset $\mathcal{S}$ with $m$ samples drawn from a distribution $\mathcal{D}$ on $\mathcal{X}_B \times \mathcal{Y}$ with $\mathcal{Y} = \{1, \ldots, d_y\}$. Given a learning algorithm (e.g., a classifier) with prior and posterior distributions $P$ and $Q$ (i.e., $\mathbf{w} + \mathbf{u}$) on the weights respectively, for any $\delta > 0$, with probability $1 - \delta$ over the draw of training data, we have that*

$$\|\mathcal{C}_{\mathcal{S}}^Q - \mathcal{C}_{\mathcal{D}}^Q\|_2 \leq \sqrt{\frac{8d_y}{m_{min} - 8d_y} \left[ D_{\mathrm{KL}}(Q\|P) + \ln\left(\frac{m_{min}}{4\delta}\right) \right]}, \tag{3}$$

*where $m_{min}$ represents the minimal number of examples from $\mathcal{S}$ which belong to the same class.*

**Adversarial setting.** Given the classifier $f_{\mathbf{w}}$ and the input data $\mathbf{x}$ with label $y$, we consider the corresponding adversarial example as

$$\mathbf{x}' = \underset{\|\mathbf{x}-\mathbf{x}'\|_p \leq \epsilon}{\arg\max} \ \ell(f_{\mathbf{w}}(\mathbf{x}'), y), \tag{4}$$

where $\epsilon$ is the $\ell_p$ norm adversarial radius. Under this adversarial setting, we let the corresponding adversarial training set and data distribution for $f_{\mathbf{w}}$ be $\mathcal{S}'$ and $\mathcal{D}'$, and adversarial confusion matrices for $f_{\mathbf{w}}$ be $\mathcal{C}_{\mathcal{D}'}^{f_{\mathbf{w}}}, \mathcal{C}_{\mathcal{S}'}^{f_{\mathbf{w}}}, \mathcal{C}_{\mathcal{S}',\gamma}^{f_{\mathbf{w}}}$. We provide more related work about adversarial training and fairness-aware adversarial learning in App. A.

**Problem definition.** For a classifier $f_{\mathbf{w}}$, the worst-class robust error is defined as

$$\max_j \ \underset{(\mathbf{x}',y)\sim\mathcal{D}'}{\mathbb{P}} (\max_{i\neq j} f_{\mathbf{w}}(\mathbf{x}')[i] \geq f_{\mathbf{w}}(\mathbf{x}')[j] \mid y = j).$$

Look at the definition in (1), one could find the sum of $j$-th column of $\mathcal{C}_{\mathcal{D}'}^{f_{\mathbf{w}}}$ represents the expected error for class $j$, which can be expressed as:

$$\sum_i (\mathcal{C}_{\mathcal{D}'}^{f_{\mathbf{w}}})_{ij} = \underset{(\mathbf{x}',y)\sim\mathcal{D}'}{\mathbb{P}} (\max_{i\neq j} f_{\mathbf{w}}(\mathbf{x}')[i] \geq f_{\mathbf{w}}(\mathbf{x}')[j] \mid y = j).$$

Since the $\ell_1$ matrix norm represents the maximum column sum of absolute elements, it naturally corresponds to the worst-class error, allowing us to express it as $\|\mathcal{C}_{\mathcal{D}'}^{f_{\mathbf{w}}}\|_1$, i.e.,

$$\|\mathcal{C}_{\mathcal{D}'}^{f_{\mathbf{w}}}\|_1 = \max_j \sum_i (\mathcal{C}_{\mathcal{D}'}^{f_{\mathbf{w}}})_{ij}.$$

In this work, we will theoretically explore how to bound $\|\mathcal{C}_{\mathcal{D}'}^{f_{\mathbf{w}}}\|_1$ in Sec. 3, and empirically study how to enhance $\|\mathcal{C}_{\mathcal{D}'}^{f_{\mathbf{w}}}\|_1$ in Sec. 4.

## 3 WORST-CLASS ROBUST ERROR AND CONFUSIONAL SPECTRAL NORM

In this section, we derive a robust generalization bound for the worst-class robust error of feed-forward neural networks with ReLU activations, leveraging the PAC-Bayesian framework. The transition from Thm. 2.1 to our theoretical results involves two key steps.

The first step is to analytically extend the PAC-Bayesian generalization bound of confusion matrices for a Gibbs classifier to a deterministic classifier. In previous works, such as Langford & Caruana (2002); Neyshabur et al. (2017a); Dziugaite & Roy (2017), leveraging PAC-Bayesian bounds to analyze the generalization behavior of neural networks focuses on evaluating the KL divergence, the perturbation error $\ell(f_{\mathbf{w}+\mathbf{u}}(\mathbf{x}), y) - \ell(f_{\mathbf{w}}(\mathbf{x}), y)$, or the numerical value of the bound. Notably, through restricting $f_{\mathbf{w}+\mathbf{u}}(\mathbf{x}) - f_{\mathbf{w}}(\mathbf{x})$, Neyshabur et al. (2017b); Farnia et al. (2019) utilize the PAC-Bayesian framework to derive margin-based bounds that are constructed by weight norms. Following these works, we adopt the margin operator with the restriction of network output. Through the eigendecomposition of the confusion matrix and the adoption of Perron–Frobenius theorem, we establish the bound between $\|\mathcal{C}_{\mathcal{D}}^{f_{\mathbf{w}}}\|_2$ and $\|\mathcal{C}_{\mathcal{S},\gamma}^{f_{\mathbf{w}}}\|_2$.

Secondly, we adapt the generalization bound from the clean setting to an adversarial context. The key idea is that the local perturbation bound in an adversarial environment can be estimated by the local perturbation bound of the clean setting, aligning with the approach in Xiao et al. (2023). The robust generalization bound retains the tightness of the standard version while incorporating an additional term $\epsilon$ that represents the perturbation radius. By leveraging the connection of $\ell_1$ matrix norm and spectral norm, we shift the bound from the overall robust accuracy to the worst-class robust accuracy. These critical steps yield the bound in Prop. 3.1. The proof is given in Sec. 3.1.

**Remark 2 (Difference with previous work)** *Building upon the key ideas introduced by Morvant et al. (2012); Neyshabur et al. (2017b); Xiao et al. (2023), we employ a chain derivation over confusion matrices to adapt previous PAC-Bayesian results to our robust generalization bound. To the best of our knowledge, this work establishes the first PAC-Bayesian robust generalization bound to characterize the worst-class performance. By extending the PAC-Bayesian framework to account for worst-class adversarial robustness, our approach provides valuable insights into the fundamental factors governing worst-class adversarial accuracy, paving the way for more effective strategies to enhance the robust fairness performance.*

**Proposition 3.1** *Consider a training set $\mathcal{S}$ with $m$ samples drawn from a distribution $\mathcal{D}$ over $\mathcal{X}_B \times \mathcal{Y}$. For any $B, n, h, \epsilon > 0$, let the base classifier $f_{\mathbf{w}} : \mathcal{X}_B \to \mathcal{Y}$ be an $n$-layer feedforward network with $h$ units each layer and ReLU activation function. Consider $\mathcal{S}'$ and $\mathcal{D}'$ as the adversarial training set and adversarial data distribution for $f_{\mathbf{w}}$ respectively, within the $\ell_2$ norm radius $\epsilon$. Then, for any $\delta, \gamma > 0$, with probability at least $1 - \delta$, we have*

$$\underbrace{\|\mathcal{C}_{\mathcal{D}'}^{f_{\mathbf{w}}}\|_1}_{\textbf{\textit{Worst-class error}}} \leq \underbrace{\nu\|\mathcal{C}_{\mathcal{S}',\gamma}^{f_{\mathbf{w}}}\|_2}_{\textbf{\textit{Empirical spectral norm}}} + \underbrace{\mathcal{O}\left(\sqrt{\frac{\nu^2 d_y}{(m_{min} - 8d_y)\gamma^2}\left[\Phi'(f_{\mathbf{w}}) + \ln\left(\frac{nm_{min}}{\delta}\right)\right]}\right)}_{\textbf{\textit{Model and training set dependence}}}, \quad (5)$$

*where $d_y$ is the number of classes, $m_{min}$ represents the minimal number of examples from $\mathcal{S}$ which belong to the same class, $\Phi'(f_{\mathbf{w}}) = (B + \epsilon)^2 n^2 h \ln(nh) \prod_{l=1}^{n} \|\mathbf{W}_l\|_2^2 \sum_{l=1}^{n} \frac{\|\mathbf{W}_l\|_F^2}{\|\mathbf{W}_l\|_2^2}$, and $\nu$ is a positive constant which depends on $d_y$.*

*Proof.* See Sec. 3.1. □

**Remark 3** *The above proposition establishes a robust generalization bound for the worst-class adversarial performance of the model, comprising two key components. The first component is the spectral norm of the empirical confusion matrix over adversarial data, while the second component is a model-dependent and training data-dependent term. The latter has been extensively studied in the previous research, with various techniques proposed to improve generalization and robust generalization, such as weight spectral norm normalization (Yoshida & Miyato, 2017; Farnia et al., 2019). In this work, from another point of the bound, we focus our attention on the spectral norm of the confusion matrix, investigating its potential to enhance the worst-class adversarial performance.*

## 3.1 SKETCH OF PROOF

Building upon the PAC-Bayesian framework established in Thm. 2.1, which bounds the gap between expected confusion matrix and empirical confusion matrix for Gibbs classifiers, our initial step is to formulate a generalization bound tailored to deterministic classifiers. By leveraging the sharpness

limit discussed in Neyshabur et al. (2017b) and employing a chain derivation of spectral norms, we have analytically derived a margin-based generalization bound for deterministic classifiers, as detailed in the following.

**Lemma 3.2** *Given Thm. 2.1, let $f_{\mathbf{w}} : \mathcal{X}_B \to \mathcal{Y}$ be any classifier with weights $\mathbf{w}$. Let $P$ be any prior distribution on the weights that is independent of the training data, $Q$ (i.e., $\mathbf{w} + \mathbf{u}$) be the posterior distribution on weights. Then, for any $\delta, \gamma > 0$, and any (posterior) random perturbation $\mathbf{u}$ s.t. $\mathbb{P}_{\mathbf{u}}(\max_{\mathbf{x}} |f_{\mathbf{w}+\mathbf{u}}(\mathbf{x}) - f_{\mathbf{w}}(\mathbf{x})|_{\infty} < \frac{\gamma}{4}) \geq \frac{1}{2}$, with probability at least $1 - \delta$, we have*

$$\|\mathcal{C}_{\mathcal{D}}^{f_{\mathbf{w}}}\|_2 \leq \|\mathcal{C}_{\mathcal{S}, \gamma}^{f_{\mathbf{w}}}\|_2 + 4\sqrt{\frac{d_y}{m_{min} - 8d_y}\left[D_{\mathrm{KL}}(\mathbf{w} + \mathbf{u}\|P) + \ln\left(\frac{3m_{min}}{4\delta}\right)\right]}. \tag{6}$$

*Proof.* See App. B. □

Although the above lemma establishes a connection between the expected and empirical confusion matrices of a deterministic classifier $f_{\mathbf{w}}$, the bound depends on the KL divergence between the posterior and prior distributions. Neyshabur et al. (2017b); Farnia et al. (2019) employ the PAC-Bayesian framework to derive a margin-based bound that depends on the weight norms through a sharpness limit, i.e., restricting the quantity $f_{\mathbf{w}+\mathbf{u}}(\mathbf{x}) - f_{\mathbf{w}}(\mathbf{x})$. Following their research, we will also replace the KL divergence term in the bound with an expression involving the weight norms.

The primary challenge lies in computing the KL divergence within the sharpness limit (or random perturbation limit), as shown in Lem. 3.2. To tackle this, we employ a two-pronged approach from Neyshabur et al. (2017b). Firstly, we leverage a pred-determined grid method to judiciously select the prior distribution $P$ of weights (classifiers). Secondly, let $\mathbf{u} \sim \mathcal{N}(0, \sigma^2\mathbf{I})$ (Neyshabur et al., 2017b), by carefully accounting for both the sharpness limit and the Lipschitz property of the model, we derive an upper bound on the randomness of posterior distribution by the weight matrices. This strategic formulation allows us to effectively bound the KL divergence between $Q$ and $P$, a crucial step in obtaining the following generalization bound.

**Lemma 3.3** *Given Lem. 3.2, for any $B, n, h > 0$, let the base classifier $f_{\mathbf{w}} : \mathcal{X}_B \to \mathcal{Y}$ be an $n$-layer feedforward network with $h$ units each layer and ReLU activation function. Then, for any $\delta, \gamma > 0$, with probability at least $1 - \delta$, we have*

$$\|\mathcal{C}_{\mathcal{D}}^{f_{\mathbf{w}}}\|_2 \leq \|\mathcal{C}_{\mathcal{S}, \gamma}^{f_{\mathbf{w}}}\|_2 + \mathcal{O}\left(\sqrt{\frac{d_y}{(m_{min} - 8d_y)\gamma^2}\left[\Phi(f_{\mathbf{w}}) + \ln\left(\frac{nm_{min}}{\delta}\right)\right]}\right), \tag{7}$$

*where $\Phi(f_{\mathbf{w}}) = B^2 n^2 h \ln(nh) \prod_{l=1}^{n} \|\mathbf{W}_l\|_2^2 \sum_{l=1}^{n} \frac{\|\mathbf{W}_l\|_F^2}{\|\mathbf{W}_l\|_2^2}$.*

*Proof.* See App. C. □

The above lemma derives a generalization guarantee by restricting the variation in the output of the network, effectively bounding the sharpness of the model through weight matrices. Building upon this foundation, we leverage a key insight from Xiao et al. (2023), specifically, that the local perturbation bounds of scalar value functions hold for both clean and adversarial settings. Capitalizing on this idea, we proceed to bound the spectral norm of the confusion matrix under adversarial perturbations to the input data. By synergistically combining the sharpness-based analysis with the adversarial perturbation framework, in the following, we establish a connection between model weights and adversarial confusion matrices. Consider the adversarial example $\mathbf{x}'$ defined in (4) for a speccific classifier $f_{\mathbf{w}}$, and the corresponding adversarial training set $\mathcal{S}'$ and adversarial data distribution $\mathcal{D}'$ for $f_{\mathbf{w}}$, we have the following lemma.

**Lemma 3.4** *Given Lem. 3.3, for any $B, n, h, \epsilon > 0$, let the base classifier $f_{\mathbf{w}} : \mathcal{X}_B \to \mathcal{Y}$ be an $n$-layer feedforward network with $h$ units each layer and ReLU activation function. Consider $\mathcal{S}'$ and $\mathcal{D}'$ as the adversarial training set and adversarial data distribution for $f_{\mathbf{w}}$ respectively, within the $\ell_2$ norm radius $\epsilon$. Then, for any $\delta, \gamma > 0$, with probability at least $1 - \delta$, we have*

$$\|\mathcal{C}_{\mathcal{D}'}^{f_{\mathbf{w}}}\|_2 \leq \|\mathcal{C}_{\mathcal{S}', \gamma}^{f_{\mathbf{w}}}\|_2 + \mathcal{O}\left(\sqrt{\frac{d_y}{(m_{min} - 8d_y)\gamma^2}\left[\Phi'(f_{\mathbf{w}}) + \ln\left(\frac{nm_{min}}{\delta}\right)\right]}\right), \tag{8}$$

*where $\Phi'(f_{\mathbf{w}}) = (B + \epsilon)^2 n^2 h \ln(nh) \prod_{l=1}^{n} \|\mathbf{W}_l\|_2^2 \sum_{l=1}^{n} \frac{\|\mathbf{W}_l\|_F^2}{\|\mathbf{W}_l\|_2^2}$.*

*Proof.* See App. D. □

The above lemma establishes a connection between the spectral norm of the expected confusion matrix, $\|\mathcal{C}_{\mathcal{D}'}^{f_\mathbf{w}}\|_2$, and the spectral norm of the empirical margin confusion matrix, $\|\mathcal{C}_{\mathcal{S}',\gamma}^{f_\mathbf{w}}\|_2$, in the adversarial setting. This connection is bridged through the information encapsulated in the training data and the model weights. Our objective, however, extends beyond this result — we aim to derive an upper bound on the worst-class adversarial performance of the model, as characterized by the $\ell_1$ norm of the expected confusion matrix under adversarial perturbations, i.e., $\|\mathcal{C}_{\mathcal{D}'}^{f_\mathbf{w}}\|_1$.

By leveraging the well-established relationship between the $\ell_1$ norm and spectral norm of matrices, we can translate the spectral norm bound obtained in the lemma into a bound on the $\ell_1$ norm, which directly governs the worst-class adversarial error of the model. Specifically, for a general confusion matrix $\mathcal{C} \in \mathbb{R}^{d_y \times d_y}$, we have the following inequality: $\|\mathcal{C}_{\mathcal{D}'}^{f_\mathbf{w}}\|_1 \leq \nu \|\mathcal{C}_{\mathcal{D}'}^{f_\mathbf{w}}\|_2$, where $\nu$ is a constant that depends on the number of classes $d_y$ and is upper bounded by $\sqrt{d_y}$. To validate the tightness of this bound, we conducted an extensive numerical study involving 1,000,000 experiments on randomly generated confusion matrices for $d_y = 10$, where the maximum of $\nu$ is 1.16 and the average value is 1.06. It demonstrates that the value of $\nu$ can be sufficiently small (i.e., close to 1), enabling the term $\nu \|\mathcal{C}_{\mathcal{D}'}^{f_\mathbf{w}}\|_2$ to provide a tight upper bound on the $\ell_1$ norm $\|\mathcal{C}_{\mathcal{D}'}^{f_\mathbf{w}}\|_1$. This numerical analysis reinforces the practical utility of our theoretical result, ensuring that the derived bound on the worst-class adversarial error is informative under PAC-Bayesian framework. Synthesizing these theoretical results and leveraging the established relationships between matrix norms, we derive our main result (Prop. 3.1) that provides a robust generalization bound for the worst-class adversarial performance of DNNs.

## 4 EMPIRICAL RESULTS

### 4.1 SPECTRAL REGULARIZATION OF CONFUSION MATRIX

Prop. 3.1 provides a theoretical perspective that identifies the spectral norm of the empirical confusion matrix as a key factor bounding the worst-class robust generalization performance. Motivated by this insight, we explore strategies to regularize this spectral term to enhance robust fairness across different classes. However, directly optimizing $\|\mathcal{C}_{\mathcal{S}',\gamma}^{f_\mathbf{w}}\|_2$ by computing the gradient with respect to the model weights presents significant challenges. As shown in (9), this difficulty arises from the discrete nature of the elements in $\mathcal{C}_{\mathcal{S}',\gamma}^{f_\mathbf{w}}$, which are binary $\{0,1\}$ error indicators. Here $(\mathcal{C}_{\mathcal{S}',\gamma}^{f_\mathbf{w}})_{ij}$ represents the element at $i$-th row, $j$-th column. The non-differentiable nature of these discrete elements precludes the straightforward application of gradient-based optimization techniques. Consequently, we are limited to employing computationally expensive methods to estimate discrete gradients, which can significantly hinder the efficiency of the regularizer. Note that in (9), for notional convenience, we use the notations including $\partial$ to represent the (discrete) gradient in practical algorithm, even though $\mathcal{C}_{\mathcal{S}',\gamma}^{f_\mathbf{w}}$ may not be theoretically differentiable.

$$\underbrace{\frac{\partial \|\mathcal{C}_{\mathcal{S}',\gamma}^{f_\mathbf{w}}\|_2}{\partial \mathbf{w}}}_{\text{Expensive}} \Longrightarrow \sum_{i \neq j} \underbrace{\frac{\partial \|\mathcal{C}_{\mathcal{S}',\gamma}^{f_\mathbf{w}}\|_2}{\partial (\mathcal{C}_{\mathcal{S}',\gamma}^{f_\mathbf{w}})_{ij}}}_{\text{Cheap}} \times \underbrace{\frac{\partial (\mathcal{C}_{\mathcal{S}',\gamma}^{f_\mathbf{w}})_{ij}}{\partial \mathbf{w}}}_{\text{Expensive}} \Longrightarrow \sum_{i \neq j} \underbrace{\frac{\partial \|\mathcal{C}_{\mathcal{S}',\gamma}^{f_\mathbf{w}}\|_2}{\partial (\mathcal{C}_{\mathcal{S}',\gamma}^{f_\mathbf{w}})_{ij}}}_{\text{Cheap}} \times \underbrace{\frac{\partial (\mathcal{C}_{\mathcal{S}',\gamma}^{f_\mathbf{w}})_{ij}}{\partial (\mathcal{L}_{\mathcal{S}',\gamma}^{f_\mathbf{w}})_{ij}}}_{\text{Approximate}} \times \underbrace{\frac{\partial (\mathcal{L}_{\mathcal{S}',\gamma}^{f_\mathbf{w}})_{ij}}{\partial \mathbf{w}}}_{\text{Cheap}} \quad (9)$$

To address the problem, as shown in the right-hand of (9), we introduce an alternative confusion matrix $\mathcal{L}_{\mathcal{S}',\gamma}^{f_\mathbf{w}}$ with differentiable elements. This matrix $\mathcal{L}_{\mathcal{S}',\gamma}^{f_\mathbf{w}}$ is structurally similar to $\mathcal{C}_{\mathcal{S}',\gamma}^{f_\mathbf{w}}$, maintaining zero diagonal elements. However, we replace the binary $\{0,1\}$ errors in the off-diagonal positions $(i,j)$ with the average KL divergence, i.e., $\forall i \neq j$,

$$(\mathcal{L}_{\mathcal{S}',\gamma}^{f_\mathbf{w}})_{ij} := \frac{1}{m_j} \sum_{(\mathbf{x}',y) \in \mathcal{S}'_{ij}} D_{\mathrm{KL}}(f_\mathbf{w}(\mathbf{x}') + \gamma(\mathbf{1} - \mathbf{y}) \| \mathbf{y}), \quad (10)$$

where $\mathbf{y}$ is the one-hot vector of $y$, $\mathcal{S}'_{ij} = \{(\mathbf{x}',y) \in \mathcal{S}' \mid f_\mathbf{w}(\mathbf{x}')[y] \leq \gamma + f_\mathbf{w}(\mathbf{x}')[i], \ y = j, \ \arg\max_{i' \neq y} f_\mathbf{w}(\mathbf{x}')[i'] = i\}$, and $m_j$ is the number of samples with label $j$ in the training set.

In the right-hand side of (9), we observe that the general direction of $\partial(\mathcal{C}_{\mathcal{S}',\gamma}^{f_\mathbf{w}})_{ij} / \partial(\mathcal{L}_{\mathcal{S}',\gamma}^{f_\mathbf{w}})_{ij}$ can be readily approximated, as the descent direction of non-diagonal elements in $\mathcal{C}_{\mathcal{S}',\gamma}^{f_\mathbf{w}}$ closely aligns

Table 1: Evaluation of different fine-tuning methods on CIFAR-10 with WRN-34-10. We report the average accuracy and the worst-class accuracy under standard $\ell_\infty$ norm Auto Attack, with baseline models of TRADES and TRADES-AWP.

| Fine-Tuning Method | Fine-Tuning Epochs | TRADES | | TRADES-AWP | |
|---|---|---|---|---|---|
| | | AA (%) | Worst (%) | AA (%) | Worst (%) |
| TRADES/AWP | - | 52.51 | 23.20 | **56.18** | 25.80 |
| + FRL-RWRM$_{0.05}$ | 80 | 49.97 | 35.40 | 46.50 | 27.70 |
| + FRL-RWRM$_{0.07}$ | 80 | 51.30 | 34.60 | 46.53 | 28.60 |
| + FAAL$_{AT}$ | 2 | 50.91 | 35.30 | 49.87 | 30.60 |
| + FAAL$_{AWP}$ | 2 | 52.45 | 35.40 | 53.93 | 37.00 |
| + Ours$_{\gamma=0.0}$ | 2 | **53.46** | **36.30** | 54.65 | 37.00 |
| + Ours$_{\gamma=0.1}$ | 2 | 53.38 | 35.60 | 54.49 | **37.60** |

Table 2: Evaluation of our fine-tuning method on different pre-trained models with DDPM generated data, over CIFAR-10 dataset and Tiny-ImageNet dataset. Our fine-tuning method is adopted on the DDPM generated training set (1M/50M) within 2 epochs.

| Dataset | Architecture | PreTrained Model Fine-Tuning Method | Clean | | AA | |
|---|---|---|---|---|---|---|
| | | | Average (%) | Worst (%) | Average (%) | Worst (%) |
| CIFAR-10 $\ell_\infty$ | WRN-28-10 (1M) | Pang et al. (2022) | **88.61** | 75.40 | 61.04 | 33.80 |
| | | + Ours$_{\gamma=0.0}$ | 88.56 | 77.30 | **61.06** | 36.30 |
| | | + Ours$_{\gamma=0.1}$ | 88.24 | **78.50** | 60.53 | **37.80** |
| | WRN-70-16 (1M) | Pang et al. (2022) | 89.01 | 75.40 | **63.34** | 35.10 |
| | | + Ours$_{\gamma=0.0}$ | **89.23** | 79.10 | 62.98 | 38.60 |
| | | + Ours$_{\gamma=0.1}$ | 88.67 | **80.20** | 62.74 | **39.30** |
| CIFAR-10 $\ell_2$ | WRN-28-10 (50M) | Wang et al. (2023) | **95.16** | 88.90 | **83.63** | 64.50 |
| | | + Ours$_{\gamma=0.0}$ | 95.02 | 89.30 | 83.60 | 66.90 |
| | | + Ours$_{\gamma=0.1}$ | 94.87 | **90.70** | 83.21 | **67.80** |
| | WRN-70-16 (50M) | Wang et al. (2023) | **95.54** | 89.30 | 84.86 | 67.00 |
| | | + Ours$_{\gamma=0.0}$ | 95.39 | 89.90 | **85.06** | 68.80 |
| | | + Ours$_{\gamma=0.1}$ | 94.98 | **91.30** | 84.20 | **69.70** |
| Tiny-ImageNet $\ell_\infty$ | WRN-28-10 (50M) | Wang et al. (2023) | **65.19** | 26.00 | 31.30 | 0.00 |
| | | + Ours$_{\gamma=0.0}$ | 65.12 | 30.00 | **31.34** | 2.00 |
| | | + Ours$_{\gamma=0.1}$ | 64.93 | **32.00** | 31.06 | **4.00** |

with that of $\mathcal{L}^{f_\mathbf{w}}_{\mathcal{S}',\gamma}$. That is, $\text{sign}\left(\frac{\partial(\mathcal{C}^{f_\mathbf{w}}_{\mathcal{S}',\gamma})_{ij}}{\partial(\mathcal{L}^{f_\mathbf{w}}_{\mathcal{S}',\gamma})_{ij}}\right) \approx 1$, where $\text{sign}(\cdot)$ denotes the sign function. This approximation strategy is well-established in machine learning; for instance, in classification tasks, while we cannot directly optimize classification error (accuracy), we optimize cross-entropy loss or KL divergence to improve accuracy based on the assumption that their optimization directions approximately coincide.

Based on this insight, we design a regularization term $\Psi(f_\mathbf{w}, \mathcal{S}', \gamma)$ as follows:

$$\frac{\partial \Psi(f_\mathbf{w}, \mathcal{S}', \gamma)}{\partial \mathbf{w}} = \sum_{i \neq j} \left\{ \frac{\partial \|\mathcal{C}^{f_\mathbf{w}}_{\mathcal{S}',\gamma}\|_2}{\partial(\mathcal{C}^{f_\mathbf{w}}_{\mathcal{S}',\gamma})_{ij}} \times \text{sign}\left(\frac{\partial(\mathcal{C}^{f_\mathbf{w}}_{\mathcal{S}',\gamma})_{ij}}{\partial(\mathcal{L}^{f_\mathbf{w}}_{\mathcal{S}',\gamma})_{ij}}\right) \times \frac{\partial(\mathcal{L}^{f_\mathbf{w}}_{\mathcal{S}',\gamma})_{ij}}{\partial \mathbf{w}} \right\}. \tag{11}$$

Incorporating this regularizer, we define the adversarial training objective function as:

$$J = \mathop{\mathbb{E}}_{(\mathbf{x}',y) \in \mathcal{S}'} \ell(f_\mathbf{w}(\mathbf{x}'), y) + \alpha \Psi(f_\mathbf{w}, \mathcal{S}', \gamma), \tag{12}$$

where $\mathcal{S}'$ is the adversarial training set for the classifier $f_\mathbf{w}$ as defined in (4), $\alpha \in (0, +\infty)$ is a hyper-parameter which balances the first adversarial training term and the second regularizer. This formulation allows us to effectively regularize the spectral norm of the confusion matrix, enabling gradient-based optimization to enhance robust fairness. In the following experiments, we normally set the default values of $\alpha$ as 0.3, $\gamma$ as 0.0 and 0.1. We provide sensitivity analysis for hyper-parameters in App. E.1.

Table 3: Training from scratch with different methods on CIFAR-10 and CIFAR-100 datasets using Preact-ResNet18 model with $\ell_\infty$ norm attack.

| AT | CIFAR-10 | | | | CIFAR-100 | | | |
|---|---|---|---|---|---|---|---|---|
| Method | Clean (%) | Worst (%) | AA (%) | Worst (%) | Clean (%) | Worst (%) | AA (%) | Worst (%) |
| PGD-AT | 82.72 | 55.80 | 47.38 | 12.90 | 54.63 | 19.00 | 22.78 | 1.00 |
| TRADES | 82.54 | 66.10 | 49.05 | 20.70 | 54.57 | 19.00 | 23.57 | 1.00 |
| CFA$_{AT}$ | 80.82 | 64.60 | **50.10** | 24.40 | 55.12 | 22.00 | 23.62 | 2.00 |
| CFA$_{TRADES}$ | 80.36 | 66.20 | **50.10** | 26.50 | 55.57 | 23.00 | 24.56 | 2.00 |
| WAT$_{TRADES}$ | 80.37 | 66.00 | 46.16 | 30.70 | 53.99 | 19.00 | 22.89 | 3.00 |
| FAAL$_{AT}$ | 82.20 | 62.90 | 49.10 | 33.70 | 56.84 | 16.00 | 21.85 | 3.00 |
| FAAL$_{TRADES}$ | 81.62 | 68.90 | 48.48 | 33.60 | 55.87 | 21.00 | 23.57 | 3.00 |
| Ours$_{\gamma=0.0}$ | **83.51** | 68.90 | 49.92 | 33.80 | **57.37** | **24.00** | **25.17** | 3.00 |
| Ours$_{\gamma=0.1}$ | 82.86 | **69.70** | 49.73 | **34.90** | 56.91 | **24.00** | 24.88 | **4.00** |

## 4.2 EXPERIMENTS

In this section, we discuss the fine-tuning performance and the adversarial training performance of our method using the average accuracy and the worst-class accuracy under AutoAttack (AA) and clean settings. AutoAttack (Croce & Hein, 2020b) is one of the most powerful attack methods, it includes three white-box attacks (APGD-CE (Croce & Hein, 2020b), APGD-DLR (Croce & Hein, 2020b), and FAB (Croce & Hein, 2020a)) and one black-box attack (Square Attack (Andriushchenko et al., 2020)). We conduct experiments on the CIFAR-10, CIFAR-100, and Tiny-ImageNet datasets, which are widely used for evaluating adversarial training methods. The perturbation budget is set to $\epsilon = 8/255$ for $\ell_\infty$ norm attacks and $\epsilon = 128/255$ for $\ell_2$ norm attacks.[1]

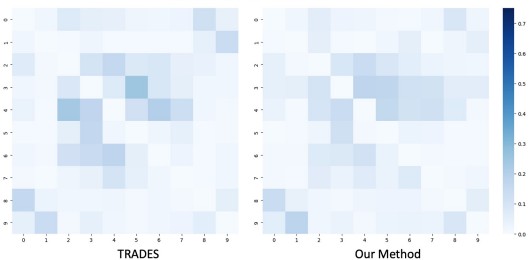

Figure 3: In both confusion matrices, the horizontal axis represents the true labels, while the vertical axis represents the predicted labels. The **left** figure shows the AA results of a WRN-34-10 model trained using the TRADES method on CIFAR-10, whereas the **right** figure demonstrates the AA results of a WRN-34-10 model trained using our method with $\gamma = 0.1$.

FRL (Xu et al., 2021) and FAAL (Zhang et al., 2024) are the existing state-of-the-art techniques from recent literature that perform fine-tuning on a pre-trained model to improve robust fairness. FRL proposes two strategies based on TRADES (Zhang et al., 2019) for enhancing robust fairness: reweight (RW) and remargin (RM). Following Zhang et al. (2024), in our experiments, we apply the best versions of FRL from their paper: FRL-RWRM with $\tau_1 = \tau_2 = 0.05$ and FRL-RWRM with $\tau_1 = \tau_2 = 0.07$, where $\tau_1$ and $\tau_2$ are the fairness constraint parameters for reweight and remargin, respectively. The results of FRL are reproduced using their public code, where the target models are fine-tuned for 80 epochs, and the best results are presented. FAAL proposes two versions based on vanilla adversarial training (AT) and AWP (Wu et al., 2020b). We provide their best results which are fine-tuned on the TRADES pre-trained WideResNet-34-10 (WRN-34-10) (Zagoruyko & Komodakis, 2016) model and the TRADES-AWP pre-trained WRN-34-10 model in Tab. 1 for CIFAR-10 with $\ell_\infty$ attack. For our method, we set the value of $\alpha$ as 0.3, $\gamma = 0.0/0.1$, and the learning rate is 0.01. As shown in Tab. 1, compared to FRL and FAAL, our method maintains higher average clean and robust accuracies while achieving better worst-class clean and robust accuracies.

Denoising Diffusion Probabilistic Model (DDPM) (Ho et al., 2020) is an advanced deep learning model primarily used for generating high-quality, diverse samples such as images or audio. The model learns to generate data by reversing a process that gradually adds noise to the data. Recently, Pang et al. (2022); Wang et al. (2023) have used DDPM-generated data to enhance adversarial training and achieved stunning results on CIFAR and Tiny-ImageNet datasets. Following their work, we fine-tune their models on CIFAR-10 under $\ell_\infty$ attack, which is pre-trained using 1M extra data; on

---

[1]https://github.com/Alexkael/CONFUSIONAL-SPECTRAL-REGULARIZATION.

Table 4: Evaluation of different fine-tuning methods on CIFAR-10 with WRN-34-10. We report the average accuracy and the worst-class accuracy under standard $\ell_\infty$ norm PGD-20/CW-20 attack, with baseline models of TRADES and TRADES-AWP.

| Fine-Tuning | TRADES | | | | TRADES-AWP | | | |
| Method | PGD-20 (%) | Worst (%) | CW-20 (%) | Worst (%) | PGD-20 (%) | Worst (%) | CW-20 (%) | Worst (%) |
|---|---|---|---|---|---|---|---|---|
| TRADES/AWP | 55.32 | 27.10 | 53.92 | 24.80 | **59.20** | 28.80 | **57.14** | 26.50 |
| + FRL-RWRM$_{0.05}$ | 53.16 | 40.60 | 51.39 | 36.30 | 49.90 | 31.70 | 49.68 | 34.00 |
| + FRL-RWRM$_{0.07}$ | 53.76 | 39.20 | 52.92 | 36.80 | 48.63 | 30.90 | 49.77 | 31.50 |
| + FAAL$_{AT}$ | 53.46 | 39.80 | 52.72 | 38.20 | 52.54 | 35.00 | 51.70 | 34.40 |
| + FAAL$_{AWP}$ | 56.07 | 43.30 | 54.16 | 38.60 | 57.14 | 43.40 | 55.34 | 40.10 |
| + Ours$_{\gamma=0.0}$ | **57.83** | **44.60** | **56.09** | **39.70** | 59.06 | 44.10 | 56.79 | 41.30 |
| + Ours$_{\gamma=0.1}$ | 57.66 | 44.10 | 55.97 | 38.90 | 58.87 | **44.50** | 56.44 | **41.80** |

Table 5: Training from scratch with different methods on CIFAR-10 using Preact-ResNet18 model with standard $\ell_2$ norm attack.

| Method | Clean (%) | Worst (%) | PGD-20 (%) | Worst (%) | CW-20 (%) | Worst (%) | AA (%) | Worst (%) |
|---|---|---|---|---|---|---|---|---|
| PGD-AT | 88.83 | 74.90 | 68.83 | 43.50 | 68.61 | 43.20 | 67.99 | 42.60 |
| FAAL$_{AT}$ | 87.61 | 76.30 | 66.57 | 46.80 | 66.31 | 46.80 | 65.45 | 44.20 |
| FAAL$_{TRADES}$ | 86.62 | 78.10 | 65.21 | 48.20 | 65.04 | 48.10 | 64.25 | 46.40 |
| Ours$_{\gamma=0.0}$ | **89.57** | **78.80** | **70.36** | 49.10 | **69.98** | 48.90 | **69.51** | 47.50 |
| Ours$_{\gamma=0.1}$ | 89.36 | 78.70 | 70.16 | **49.90** | 69.74 | **49.60** | 69.32 | **48.40** |

CIFAR-10 under $\ell_2$ attack, which is pre-trained using 50M extra data; and on Tiny-ImageNet under $\ell_\infty$ attack, which is pre-trained using 50M extra data. Our fine-tuning setting is the same as in the previous experiment. As shown in Tab. 2, after fine-tuning within 2 epochs, our method significantly improves worst-class clean and robust accuracies while maintaining the average accuracy or only experiencing a small decline in average accuracy.

We also investigate the advancements achieved by training the model from the ground up using our method. We compare our methods with two common adversarial training methods, PGD-AT and TRADES, and three recent state-of-the-art techniques: CFA (Wei et al., 2023), WAT (Li & Liu, 2023), and FAAL, which have been recently proposed to mitigate robust fairness issues. Note that our method builds upon the TRADES adversarial training framework here. We adversarially trained Preact-ResNet-18 models (He et al., 2016) for 200 epochs using SGD with a momentum of 0.9, batch size of 128, weight decay of $5 \times 10^{-4}$, and an initial learning rate of 0.1, which is reduced by a factor of 10 at the 100th and 150th epochs. Following Zhang et al. (2024), we report the best results under Auto Attack for both average accuracy and worst-class accuracy in Tab. 3. Furthermore, as shown in Fig. 3, the confusion matrix generated by WRN-34-10 model trained with our method (right) exhibits a more fair (uniform) distribution compared with the one generated by TRADES (left). Here we adopt the confusion matrix defined in (1), where the diagonal elements are 0.

Additionally, we extend our evaluation to include fine-tuning experiments under both standard $\ell_\infty$ norm PGD-20 attack and CW-20 attack (Carlini & Wagner, 2017). As demonstrated in Tab. 4, our method maintains superior worst-class performance across these different attack types. Furthermore, we evaluate models trained from scratch under $\ell_2$ norm attacks. The results in Tab. 5 show that for CIFAR-10 with Preact-ResNet18 architecture, our method achieves the best performance in both average and worst-class robust accuracy under standard $\ell_2$ attacks.

## 5  CONCLUSION

This work addresses an oversight in the robust fairness literature by arguing that the spectral norm of the confusion matrix over training data needs to be systematically considered. Through theoretical study (updating the PAC-Bayesian framework), algorithmic development (efficient regularization of the confusional spectral norm), and extensive experiments, we demonstrate that the consideration of the spectral norm of the confusion matrix can improve the worst-class robust performance and robust fairness over not only the vanilla adversarial training framework but also the state-of-the-art adversarially trained models.

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

# A    RELATED WORK

**Adversarial training** (Engstrom et al., 2018; Kannan et al., 2018; Zhang et al., 2020; Lee et al., 2020; Huang et al., 2023; 2021) is one of the most effective methods against adversarial attacks (Lin et al., 2024), normally, it can be formulated as a minimax optimization problem (Madry et al., 2018)

$$\min_{\mathbf{w}} \left\{ \mathbb{E}_{(\mathbf{x}',y)\sim\mathcal{S}'} \left[ \ell(f_{\mathbf{w}}(\mathbf{x}'), y) \right] \right\}, \tag{13}$$

where $\mathbf{x}'$ is an adversarial example causing the largest loss for $f_{\mathbf{w}}$ within an $\epsilon$-ball as defined in (4).

Recently, several works (Li & Liu, 2023; Ma et al., 2022; Sun et al., 2023; Wei et al., 2023; Xu et al., 2021; Zhang et al., 2024) have explored ways to address the **fairness issue in adversarial robustness** (the imbalance issue for robust accuracies across classes). Xu et al. (2021) was the first to reveal that the issue of robust fairness occurs in conventional adversarial training, which can introduce severe disparity in accuracy and robustness between different groups of data when boosting the average robustness. To mitigate this problem, they proposed a Fair-Robust-Learning (FRL) framework that employs reweight and remargin strategies to fine-tune the pre-trained model, reducing the significant boundary error within a certain margin. Ma et al. (2022) empirically discovered that a trade-off exists between robustness and robustness fairness, and adversarial training (AT) with a larger perturbation radius will result in a larger variance. To mitigate this trade-off, they added a variance regularization term to the objective function, naming the method FAT (Fairness-Aware Adversarial Training), which relieves the trade-off between average robustness and robust fairness. Sun et al. (2023) proposed a method called Balance Adversarial Training (BAT), which adjusts the attack strengths and difficulties of each class to generate samples near the decision boundary for easier and fairer model learning. Wei et al. (2023) presented a framework named CFA (Class-wise Fair Adversarial training), which automatically customizes specific training configurations for each class, thereby improving the worst-class robustness while maintaining the average performance. More recently, Li & Liu (2023) considered the worst-class robust risk and proposed a framework named WAT (Worst-class Adversarial Training), leveraging no-regret dynamics to solve this problem. Zhang et al. (2024) proposed Fairness-Aware Adversarial Learning (FAAL) to enhance robust fairness by considering the worst-case distribution across various classes.

Unlike the above works, this study develops a robust generalization bound for the worst-class robust error and proposes a method to enhance worst-class robust performance and robust fairness by regularizing the spectral norm of the robust confusion matrix.

# B PROOF OF LEM. 3.2

**Proof B.1** *Let $\mathcal{S}_{\mathbf{u}}$ be the set of perturbations with the following property:*

$$\mathcal{S}_{\mathbf{u}} \subseteq \left\{ \mathbf{u} \,\Big|\, \max_{\mathbf{x} \in \mathcal{X}_B} |f_{\mathbf{w}+\mathbf{u}}(\mathbf{x}) - f_{\mathbf{w}}(\mathbf{x})|_{\infty} < \frac{\gamma}{4} \right\}. \tag{14}$$

*Let $q$ be the probability density function over $\mathbf{u}$. We construct a new distribution $\tilde{Q}$ over $\tilde{\mathbf{u}}$ that is restricted to $\mathcal{S}_{\mathbf{u}}$ with the probability density function:*

$$\tilde{q}(\tilde{\mathbf{u}}) = \begin{cases} \frac{1}{z}q(\tilde{\mathbf{u}}) & \tilde{\mathbf{u}} \in \mathcal{S}_{\mathbf{u}}, \\ 0 & otherwise, \end{cases} \tag{15}$$

*where $z$ is a normalizing constant and by the lemma assumption $z = \mathbb{P}(\mathbf{u} \in \mathcal{S}_{\mathbf{u}}) \geq \frac{1}{2}$. By the definition of $\tilde{Q}$, we have:*

$$\max_{\mathbf{x} \in \mathcal{X}_B} |f_{\mathbf{w}+\tilde{\mathbf{u}}}(\mathbf{x}) - f_{\mathbf{w}}(\mathbf{x})|_{\infty} < \frac{\gamma}{4}. \tag{16}$$

*Therefore, with probability at least $1 - \delta$ over training dataset $\mathcal{S}$, we have:*

$$\|\mathcal{C}_{\mathcal{D}}^{f_{\mathbf{w}}}\|_2 \leq \|\mathcal{C}_{\mathcal{D},\frac{\gamma}{2}}^{\tilde{Q}}\|_2 \qquad \qquad \triangleright because\ of\ Pf.\ B.2$$

$$\leq \|\mathcal{C}_{\mathcal{S},\frac{\gamma}{2}}^{\tilde{Q}}\|_2 + \sqrt{\frac{8d_y}{m_{min} - 8d_y}\left[D_{\mathrm{KL}}(\tilde{Q}\|P) + \ln\left(\frac{m_{min}}{4\delta}\right)\right]} \qquad \triangleright because\ of\ Pf.\ B.3$$

$$\leq \|\mathcal{C}_{\mathcal{S},\gamma}^{f_{\mathbf{w}}}\|_2 + \sqrt{\frac{8d_y}{m_{min} - 8d_y}\left[D_{\mathrm{KL}}(\tilde{Q}\|P) + \ln\left(\frac{m_{min}}{4\delta}\right)\right]} \qquad \triangleright because\ of\ Pf.\ B.4$$

$$\leq \|\mathcal{C}_{\mathcal{S},\gamma}^{f_{\mathbf{w}}}\|_2 + 4\sqrt{\frac{d_y}{m_{min} - 8d_y}\left[D_{\mathrm{KL}}(Q\|P) + \ln\left(\frac{3m_{min}}{4\delta}\right)\right]} \qquad \triangleright because\ of\ Pf.\ B.5$$

*Hence, proved.* $\qquad\qquad\square$

**Proof B.2** *Given (14) and (15), for all $\tilde{\mathbf{u}} \in \tilde{Q}$, we have*

$$\max_{\mathbf{x} \in \mathcal{X}_B} |f_{\mathbf{w}+\tilde{\mathbf{u}}}(\mathbf{x}) - f_{\mathbf{w}}(\mathbf{x})|_{\infty} < \frac{\gamma}{4}. \tag{17}$$

*For all $\mathbf{x} \in \mathcal{X}_B$ s.t. $\arg\max_i f_{\mathbf{w}}(\mathbf{x})[i] \neq y$, we have*

$$f_{\mathbf{w}+\tilde{\mathbf{u}}}(\mathbf{x})[\arg\max_i f_{\mathbf{w}}(\mathbf{x})[i]] + \frac{\gamma}{4} \geq f_{\mathbf{w}+\tilde{\mathbf{u}}}(\mathbf{x})[y] - \frac{\gamma}{4}. \tag{18}$$

*Thus for all $i \neq j$, we have*

$$(\mathcal{C}_{\mathcal{D}}^{f_{\mathbf{w}}})_{ij} \leq (\mathcal{C}_{\mathcal{D},\frac{\gamma}{2}}^{\tilde{Q}})_{ij}. \tag{19}$$

*According to Perron–Frobenius theorem (Frobenius et al., 1912), for all $1 \leq i, j \leq d_y$, $\frac{\partial \|\mathcal{C}\|_2}{\partial(\mathcal{C})_{ij}} \geq 0$. Combine the above conditions, we get $\|\mathcal{C}_{\mathcal{D}}^{f_{\mathbf{w}}}\|_2 \leq \|\mathcal{C}_{\mathcal{D},\frac{\gamma}{2}}^{\tilde{Q}}\|_2$.* $\qquad\square$

**Proof B.3** *Apply $|\|\mathbf{A}\|_2 - \|\mathbf{B}\|_2| \leq \|\mathbf{A} - \mathbf{B}\|_2$ and Thm. 2.1.* $\qquad\square$

**Proof B.4** *For all $\mathbf{x} \in \mathcal{X}_B$, if there exists $\tilde{\mathbf{u}} \in \tilde{Q}$ s.t. $\max_{i \neq y} f_{\mathbf{w}+\tilde{\mathbf{u}}}(\mathbf{x})[i] + \frac{\gamma}{2} \geq f_{\mathbf{w}+\tilde{\mathbf{u}}}(\mathbf{x})[y]$, we have*

$$\max_{i \neq y} f_{\mathbf{w}}(\mathbf{x})[i] + \gamma \geq f_{\mathbf{w}}(\mathbf{x})[y]. \tag{20}$$

*Thus for all $i \neq j$, we have*

$$(\mathcal{C}_{\mathcal{S},\frac{\gamma}{2}}^{\tilde{Q}})_{ij} \leq (\mathcal{C}_{\mathcal{S},\gamma}^{f_{\mathbf{w}}})_{ij}. \tag{21}$$

*According to Perron–Frobenius theorem, for all $1 \leq i, j \leq d_y$, $\frac{\partial \|\mathcal{C}\|_2}{\partial(\mathcal{C})_{ij}} \geq 0$. Combine the above conditions, we get $\|\mathcal{C}_{\mathcal{S},\frac{\gamma}{2}}^{\tilde{Q}}\|_2 \leq \|\mathcal{C}_{\mathcal{S},\gamma}^{f_{\mathbf{w}}}\|_2$.*

**Proof B.5** *Given $q$, $\tilde{q}$, $z$, and $\mathcal{S}_{\mathbf{u}}$ in (15), let $\mathcal{S}_{\mathbf{u}}^c$ denote the complement set of $\mathcal{S}_{\mathbf{u}}$ and $\tilde{q}^c$ denote the normalized density function restricted to $\mathcal{S}_{\mathbf{u}}^c$. Then, we have*

$$D_{\mathrm{KL}}(q\|p) = zD_{\mathrm{KL}}(\tilde{q}\|p) + (1-z)D_{\mathrm{KL}}(\tilde{q}^c\|p) - H(z), \tag{22}$$

*where $H(z) = -z\ln z - (1-z)\ln(1-z) \leq 1$ is the binary entropy function. Since $D_{\mathrm{KL}}$ is always positive, we get*

$$D_{\mathrm{KL}}(\tilde{q}\|p) = \frac{1}{z}[D_{\mathrm{KL}}(q\|p) + H(z)) - (1-z)D_{\mathrm{KL}}(\tilde{q}^c\|p)] \leq 2(D_{\mathrm{KL}}(q\|p) + 1). \tag{23}$$

*Thus we have $2(D_{\mathrm{KL}}(\mathbf{w} + \mathbf{u}\|P) + \ln\frac{3m_{min}}{4\delta}) \geq D_{\mathrm{KL}}(\mathbf{w} + \tilde{\mathbf{u}}\|P) + \ln\frac{m_{min}}{4\delta}$.* □

## C  PROOF OF LEM. 3.3

**Proof C.1** *Following Neyshabur et al. (2017b), we use two main steps to prove Lem. 3.3. Firstly, we compute the maximum allowable perturbation of $\mathbf{u}$ required to satisfy the given condition on the margin $\gamma$. In the second step, we compute the KL term in the bound, considering the perturbation obtained from the previous step. This computation is essential in deriving the PAC-Bayesian bound.*

*Consider a neural network with weights $\mathbf{W}$ that can be regularized by dividing each weight matrix $\mathbf{W}_l$ by its spectral norm $\|\mathbf{W}_l\|_2$. Let $\beta$ be the geometric mean of the spectral norms of all weight matrices, defined as:*

$$\beta = \left(\prod_{l=1}^n \|\mathbf{W}_l\|_2\right)^{\frac{1}{n}},$$

*where $n$ is the number of weight matrices in the network. We introduce a modified version of the weights, denoted as $\widetilde{\mathbf{W}}_l$, which is obtained by scaling the original weights $\mathbf{W}_l$ by a factor of $\frac{\beta}{\|\mathbf{W}_l\|_2}$:*

$$\widetilde{\mathbf{W}}_l = \frac{\beta}{\|\mathbf{W}_l\|_2}\mathbf{W}_l.$$

*Due to the homogeneity property of the ReLU activation function, the behavior of the network with the modified weights, denoted as $f_{\widetilde{\mathbf{w}}}$, is identical to that of the original network $f_{\mathbf{w}}$.*

*Furthermore, we observe that the product of the spectral norms of the original weights, given by $\prod_{l=1}^n \|\mathbf{W}_l\|_2$, is equal to the product of the spectral norms of the modified weights, expressed as $\prod_{l=1}^n \|\widetilde{\mathbf{W}}_l\|_2$. Moreover, the ratio of the Frobenius norm to the spectral norm remains unchanged for both the original and modified weights:*

$$\frac{\|\mathbf{W}_l\|_F}{\|\mathbf{W}_l\|_2} = \frac{\|\widetilde{\mathbf{W}}_l\|_F}{\|\widetilde{\mathbf{W}}_l\|_2}.$$

*As a result, the excess error mentioned in the theorem statement remains unaffected by this weight normalization. Therefore, it is sufficient to prove the theorem only for the normalized weights $\widetilde{\mathbf{w}}$. Without loss of generality, we assume that the spectral norm of each weight matrix is equal to $\beta$, i.e., $\|\mathbf{W}_l\|_2 = \beta$ for any layer $l$.*

*In our approach, we initially set the prior distribution $P$ as a Gaussian distribution with zero mean and a diagonal covariance matrix $\sigma^2\mathbf{I}$. We incorporate random perturbations $\mathbf{u} \sim \mathcal{N}(0, \sigma^2\mathbf{I})$, where the value of $\sigma$ will be determined in relation to $\beta$ at a later stage. Since the prior must be independent of the learned predictor $\mathbf{w}$ and its norm, we choose $\sigma$ according to an estimated value $\tilde{\beta}$. We calculate the PAC-Bayesian bound for each $\tilde{\beta}$ selected from a pre-determined grid, offering a generalization guarantee for all $\mathbf{w}$ satisfying $|\beta - \tilde{\beta}| \leq \frac{1}{n}\beta$. This ensures that each relevant $\beta$ value is covered by some $\tilde{\beta}$ in the grid. Subsequently, we apply a union bound across all $\tilde{\beta}$ defined by the grid. For now, we will consider a set of $\tilde{\beta}$ and the corresponding $\mathbf{w}$ that meet the condition $|\beta - \tilde{\beta}| \leq \frac{1}{n}\beta$, which implies:*

$$\frac{1}{e}\beta^{n-1} \leq \tilde{\beta}^{n-1} \leq e\beta^{n-1}.$$

*According to Bandeira & Boedihardjo (2021) and the fact that $\mathbf{u} \sim \mathcal{N}(0, \sigma^2 \mathbf{I})$, we can obtain the following bound for the spectral norm of the perturbation matrix $\mathbf{U}_l$ ($\mathbf{u}_l = vec(\mathbf{U}_l)$):*

$$\mathbb{P}_{\mathbf{u}_l \sim \mathcal{N}(0, \sigma^2 \mathbf{I})} \left[ \|\mathbf{U}_l\|_2 > t \right] \leq 2h \exp\left( -\frac{t^2}{2h\sigma^2} \right), \tag{24}$$

*where $h$ is the width of the hidden layers. By taking a union bound over the layers, we can establish that, with a probability of at least $\frac{1}{2}$, the spectral norm of the perturbation $\mathbf{U}_l$ in each layer is bounded by $\sigma\sqrt{2h\ln(4nh)}$.*

*Plugging the bound into Lem. C.2, we have that*

$$\begin{aligned}
\max_{\mathbf{x} \in \mathcal{X}_B} \|f_{\mathbf{w}+\mathbf{u}}(\mathbf{x}) - f_{\mathbf{w}}(\mathbf{x})\|_2 &\leq eB\beta^n \sum_l \frac{\|\mathbf{U}_l\|_2}{\beta} \\
&= eB\beta^{n-1} \sum_l \|\mathbf{U}_l\|_2 \\
&\leq e^2 nB\tilde{\beta}^{n-1}\sigma\sqrt{2h\ln(4nh)} \leq \frac{\gamma}{4}.
\end{aligned} \tag{25}$$

*To make (25) hold, given $\tilde{\beta}^{n-1} \leq e\beta^{n-1}$, we can choose the largest $\sigma$ as*

$$\sigma = \frac{\gamma}{114nB\sqrt{h\ln(4nh)} \prod_{l=1}^n \|\mathbf{W}_l\|_2^{\frac{n-1}{n}}}.$$

*Hence, the perturbation $\mathbf{u}$ with the above value of $\sigma$ satisfies the assumptions of the Lem. 3.2. We now compute the KL-term using the selected distributions for $P$ and $Q$, considering the given value of $\sigma$,*

$$\begin{aligned}
D_{\mathrm{KL}}(\mathbf{w}+\mathbf{u}\|P) &\leq \frac{\|\mathbf{w}\|_2^2}{2\sigma^2} \\
&= \frac{\sum_{l=1}^n \|\mathbf{W}_l\|_F^2}{2\sigma^2} \\
&\leq \mathcal{O}\left( B^2 n^2 h \ln(nh) \frac{\prod_{l=1}^n \|\mathbf{W}_l\|_2^2}{\gamma^2} \sum_{l=1}^n \frac{\|\mathbf{W}_l\|_F^2}{\|\mathbf{W}_l\|_2^2} \right).
\end{aligned}$$

*Then, we can give a union bound over different choices of $\tilde{\beta}$. We only need to form the bound for $\left(\frac{\gamma}{2B}\right)^{\frac{1}{n}} \leq \beta \leq \left(\frac{\gamma\sqrt{m}}{2B}\right)^{\frac{1}{n}}$ which can be covered using a cover of size $nm^{\frac{1}{2n}}$ as discussed in Neyshabur et al. (2017b). Thus, with probability $\geq 1 - \delta$, for any $\tilde{\beta}$ and for all $\mathbf{w}$ such that $|\beta - \tilde{\beta}| \leq \frac{1}{n}\beta$, we have:*

$$\|\mathcal{C}_{\mathcal{D}}^{f_{\mathbf{w}}}\|_2 \leq \|\mathcal{C}_{\mathcal{S},\gamma}^{f_{\mathbf{w}}}\|_2 + \mathcal{O}\left( \sqrt{\frac{d_y}{(m_{min} - 8d_y)\gamma^2} \left[ \Phi(f_{\mathbf{w}}) + \ln\left(\frac{nm_{min}}{\delta}\right) \right]} \right), \tag{26}$$

*where $\Phi(f_{\mathbf{w}}) = B^2 n^2 h \ln(nh) \prod_{l=1}^n \|\mathbf{W}_l\|_2^2 \sum_{l=1}^n \frac{\|\mathbf{W}_l\|_F^2}{\|\mathbf{W}_l\|_2^2}$.*

*Hence, proved.* $\qquad\square$

**Lemma C.2 (Neyshabur et al. (2017b))** *For any $B, n > 0$, let $f_{\mathbf{w}} : \mathcal{X}_B \rightarrow \mathcal{Y}$ be a $n$-layer feedforward network with ReLU activation function. Then for any $\mathbf{w}$, and $\mathbf{x} \in \mathcal{X}_B$, and any perturbation $\mathbf{u} = vec(\{\mathbf{U}_l\}_{l=1}^n)$ such that $\|\mathbf{U}_l\|_2 \leq \frac{1}{n}\|\mathbf{W}_l\|_2$, the change in the output of the network can be bounded as follow*

$$\|f_{\mathbf{w}+\mathbf{u}}(\mathbf{x}) - f_{\mathbf{w}}(\mathbf{x})\|_2 \leq eB\left( \prod_{l=1}^n \|\mathbf{W}_l\|_2 \right) \sum_{l=1}^n \frac{\|\mathbf{U}_l\|_2}{\|\mathbf{W}_l\|_2}. \tag{27}$$

# D   PROOF FOR LEM. 3.4

**Proof D.1** *Following the proof process from Xiao et al. (2023), we first introduce some definitions. Then, we derive Lems. D.4 and D.5. By combining these results, we obtain Lem. 3.4. Note that we provide a concise process here, a more detailed one can be found in Xiao et al. (2023).*

**Definition D.2 (Local perturbation bound)** *Given $\mathbf{x} \in \mathcal{X}_B$, we say $g_{\mathbf{w}}(\mathbf{x})$ has a $(L_1, \cdots, L_n)$-local perturbation bound w.r.t. $\mathbf{w}$, if*

$$|g_{\mathbf{w}}(\mathbf{x}) - g_{\mathbf{w}'}(\mathbf{x})| \leq \sum_{l=1}^{n} L_l \|\mathbf{W}_l - \mathbf{W}'_l\|_2,$$

*where $L_l$ can be related to $\mathbf{w}, \mathbf{w}'$ and $\mathbf{x}$.*

*The bound introduced in Def. D.2 plays a crucial role in quantifying the variation in the output of the function $g_{\mathbf{w}}(\mathbf{x})$, especially when the weights are subject to small perturbations. Building upon this foundation, we can derive the following lemma, as demonstrated in the work of Xiao et al. (2023).*

**Lemma D.3 (Xiao et al. (2023))** *If $g_{\mathbf{w}}(\mathbf{x})$ has a $(A_1|\mathbf{x}|, \cdots, A_n|\mathbf{x}|)$-local perturbation bound, i.e.,*

$$|g_{\mathbf{w}}(\mathbf{x}) - g_{\mathbf{w}'}(\mathbf{x})| \leq \sum_{l=1}^{n} A_l |\mathbf{x}| \|\mathbf{W}_l - \mathbf{W}'_l\|_2,$$

*the robustified function $\max_{\|\mathbf{x}-\mathbf{x}'\|_2 \leq \epsilon} g_{\mathbf{w}}(\mathbf{x}')$ has a $(A_1(|\mathbf{x}| + \epsilon), \cdots, A_n(|\mathbf{x}| + \epsilon))$-local perturbation bound.*

**Margin Operator.** *Following the notation used by Bartlett et al. (2017); Xiao et al. (2023), the margin operator is defined for both the true label $y$ given an input $\mathbf{x}$ and for a pair of classes $(i, j)$. This definition provides a clear and precise measure of class separation for the model, quantifying the difference between the model's output for the true class and other classes, as well as the difference between the model's outputs for any pair of classes.*

$$\begin{aligned} M(f_{\mathbf{w}}(\mathbf{x}), y) &= f_{\mathbf{w}}(\mathbf{x})[y] - \max_{i \neq y} f_{\mathbf{w}}(\mathbf{x})[i], \\ M(f_{\mathbf{w}}(\mathbf{x}), i, j) &= f_{\mathbf{w}}(\mathbf{x})[i] - f_{\mathbf{w}}(\mathbf{x})[j]. \end{aligned} \tag{28}$$

**Robust Margin Operator.** *Similarly, the robust margin operator is also defined for a pair of classes $(i, j)$ with respect to $(\mathbf{x}, y)$.*

$$\begin{aligned} RM(f_{\mathbf{w}}(\mathbf{x}), y) &= \max_{\|\mathbf{x}-\mathbf{x}'\|_2 \leq \epsilon} \left( f_{\mathbf{w}}(\mathbf{x}')[y] - \max_{j \neq y} f_{\mathbf{w}}(\mathbf{x}')[j] \right), \\ RM(f_{\mathbf{w}}(\mathbf{x}), i, j) &= \max_{\|\mathbf{x}-\mathbf{x}'\|_2 \leq \epsilon} \left( f_{\mathbf{w}}(\mathbf{x}')[i] - f_{\mathbf{w}}(\mathbf{x}')[j] \right). \end{aligned} \tag{29}$$

*Building upon the previously introduced definitions and Lem. D.3, Xiao et al. (2023) presents the form of $A_i$ for the margin operator in the following lemma.*

**Lemma D.4 (Xiao et al. (2023))** *Consider $f_{\mathbf{w}}(\cdot)$ as an $n$-layer neural network characterized by ReLU activation functions. It is established that the following bounds apply to local perturbations within this framework.*

*1. Given $\mathbf{x}$ and $i, j$, the margin operator $M(f_{\mathbf{w}}(\mathbf{x}), i, j)$ has a $(A_1|\mathbf{x}|, , A_n|\mathbf{x}|)$-local perturbation bound w.r.t. $\mathbf{w}$, where $A_l = 2e \prod_{l=1}^{n} \|\mathbf{W}_l\|_2 / \|\mathbf{W}_l\|_2$. And*

$$|M(f_{\mathbf{w}+\mathbf{u}}(\mathbf{x}), i, j) - M(f_{\mathbf{w}}(\mathbf{x}), i, j)| \leq 2eB \prod_{l=1}^{n} \|\mathbf{W}_l\|_2 \sum_{l=1}^{n} \frac{\|\mathbf{U}_l\|_2}{\|\mathbf{W}_l\|_2}. \tag{30}$$

*2. Given $\mathbf{x}$ and $i, j$, the robust margin operator $RM(f_{\mathbf{w}}(\mathbf{x}), i, j)$ has a $(A_1(|\mathbf{x}| + \epsilon), , A_n(|\mathbf{x}| + \epsilon))$-local perturbation bound w.r.t. $\mathbf{w}$. And*

$$|RM(f_{\mathbf{w}+\mathbf{u}}(\mathbf{x}), i, j) - RM(f_{\mathbf{w}}(\mathbf{x}), i, j)| \leq 2e(B + \epsilon) \prod_{l=1}^{n} \|\mathbf{W}_l\|_2 \sum_{l=1}^{n} \frac{\|\mathbf{U}_l\|_2}{\|\mathbf{W}_l\|_2}. \tag{31}$$

*By combining Lems. 3.2 and D.4, and building upon the work of Xiao et al. (2023), we obtain the following lemma, which demonstrates that the weight perturbation of the robust margin operator can be utilized to develop a PAC-Bayesian framework.*

**Lemma D.5** *Let $f_{\mathbf{w}} : \mathcal{X}_B \to \mathbb{R}^{n_y}$ be any predictor with weights $\mathbf{w}$, and $P$ be any distribution on the weights that is independent of the training data. Then, for any $\gamma, \delta > 0$, with probability at least $1 - \delta$ over the training set of size $m$, for any $\mathbf{w}$, and any random perturbation $\mathbf{u}$ s.t.*

*1.*

$$\mathbb{P}_{\mathbf{u}} \left[ \max_{i,j \in [d^y], \mathbf{x} \in \mathcal{X}_B} |M(f_{\mathbf{w}+\mathbf{u}}(\mathbf{x}), i, j) - M(f_{\mathbf{w}}(\mathbf{x}), i, j)| \leq \frac{\gamma}{2} \right] \geq \frac{1}{2},$$

*we have*

$$\|\mathcal{C}_{\mathcal{D}}^{f_{\mathbf{w}}}\|_2 \leq \|\mathcal{C}_{\mathcal{S},\gamma}^{f_{\mathbf{w}}}\|_2 + 4\sqrt{\frac{d_y}{m_{min} - 8d_y} \left[ D_{\mathrm{KL}}(\mathbf{w} + \mathbf{u}\|P) + \ln\left(\frac{3m_{min}}{4\delta}\right) \right]}.$$

*2.*

$$\mathbb{P}_{\mathbf{u}} \left[ \max_{i,j \in [d^y], \mathbf{x} \in \mathcal{X}_B} |RM(f_{\mathbf{w}+\mathbf{u}}(\mathbf{x}), i, j) - RM(f_{\mathbf{w}}(\mathbf{x}), i, j)| \leq \frac{\gamma}{2} \right] \geq \frac{1}{2},$$

*we have*

$$\|\mathcal{C}_{\mathcal{D}'}^{f_{\mathbf{w}}}\|_2 \leq \|\mathcal{C}_{\mathcal{S}',\gamma}^{f_{\mathbf{w}}}\|_2 + 4\sqrt{\frac{d_y}{m_{min} - 8d_y} \left[ D_{\mathrm{KL}}(\mathbf{w} + \mathbf{u}\|P) + \ln\left(\frac{3m_{min}}{4\delta}\right) \right]}. \tag{32}$$

*Finally, combine Lems. D.4, D.5 and Lem. 3.3 (App. C), we get Lem. 3.4.* □

# E MORE EXPERIMENTAL RESULTS

## E.1 SENSITIVITY ANALYSIS FOR HYPER-PARAMETERS

Table 6: Training from scratch on CIFAR-10 using Preact-ResNet18 model under $\ell_\infty$ norm attack with different hyper-parameters. Left table shows results for fixed $\gamma = 0.0$ with $\alpha$ ranging from 0.0 to 0.4. Right table shows results for fixed $\alpha = 0.3$ with $\gamma$ ranging from 0.0 to 0.4.

| | $\gamma = 0.0$ | | | $\alpha = 0.3$ | |
|---|---|---|---|---|---|
| $\alpha$ | AA (%) | Worst (%) | $\gamma$ | AA (%) | Worst (%) |
| 0.0 | 47.38 | 12.90 | 0.0 | **49.92** | 33.80 |
| 0.1 | 49.02 | 21.50 | 0.1 | 49.73 | **34.90** |
| 0.2 | 49.76 | 24.40 | 0.2 | 49.31 | 34.70 |
| 0.3 | **49.92** | **33.80** | 0.3 | 48.56 | 33.90 |
| 0.4 | 49.13 | 30.10 | 0.4 | 47.68 | 33.20 |

We investigate how regularization hyper-parameters $\alpha$ and $\gamma$ influence model performance through experiments on CIFAR-10 with PreAct ResNet-18. Models are trained for 200 epochs with batch size 128, using two configurations: (1) fixed $\gamma = 0.0$ with $\alpha$ varying from 0.0 to 0.4, and (2) fixed $\alpha = 0.3$ with $\gamma$ varying from 0.0 to 0.4. As shown in Tab. 6 (left), both average and worst-class AA accuracy initially increase with $\alpha$ before declining due to the disruption of training performance at larger values, leading to our choice of $\alpha = 0.3$ for subsequent experiments. Tab. 6 (right) demonstrates that with increasing $\gamma$, average AA accuracy consistently decreases while worst-class AA accuracy shows an initial improvement followed by deterioration. This behavior can be explained by two competing effects: while larger $\gamma$ values may impair overall training performance, they also reduce the influence of the final term in Prop. 3.1, allowing the regularization term to more effectively optimize worst-class AA accuracy. This trade-off explains why $\gamma = 0.1$ achieves better worst-class performance while maintaining acceptable average performance.

Table 7: Following Cui et al. (2024), we conduct experiments on ImageNet and CIFAR-100 with clean training, compare ours with other methods on fairness.

| Dataset | Method | Easy (%) | Medium (%) | Hard (%) | All (%) |
|---|---|---|---|---|---|
| ImageNet | Baseline (ResNet-50) | 93.1 | 81.1 | 59.4 | 77.8 |
| | Menon et al. (2021) | 91.7(-1.4) | 79.8(-1.3) | 61.4(+2.0) | 77.6 |
| | Cui et al. (2019) | 91.6(-1.5) | 79.6(-1.5) | 61.3(+1.9) | 77.5 |
| | Ours$_{\gamma=0.0}$ | 91.5(-1.6) | 79.6(-1.5) | **62.3(+2.9)** | 77.8 |
| CIFAR-100 | Baseline (WRN-34-10) | 92.2 | 83.2 | 70.1 | 81.7 |
| | Menon et al. (2021) | 91.5(-0.7) | 83.2(+0.0) | 70.4(+0.3) | 81.6 |
| | Nam et al. (2020) | 90.4(-1.8) | 81.5(-1.7) | 67.7(-2.4) | 79.7 |
| | Liu et al. (2021) | 91.3(-0.9) | 82.4(-0.8) | 69.6(-0.5) | 81.0 |
| | Ours$_{\gamma=0.0}$ | 91.3(-0.9) | 83.0(-0.2) | **71.2(+1.1)** | 81.8 |

### E.2 EXPERIMENTS ON CLEAN TRAINING

Following Cui et al. (2024), we extend our evaluation to clean training scenarios on ImageNet and CIFAR-100, comparing our method with other approaches for clean accuracy fairness. As demonstrated in Tab. 7, our method effectively enhances fairness in clean training, improving test accuracy on "hard" subsets of ImageNet and CIFAR-100 while maintaining overall accuracy.

### E.3 SHARPNESS ANALYSIS

Table 8: Sharpness-like method estimated variance with respect to robust generalization gap and worst-class AA accuracy. The models are trained from scratch on CIFAR-10 using Preact-ResNet18 model with $\ell_\infty$ norm attack.

| Method | Training AA - Test AA (%) | Test worst-class AA (%) | Largest variance |
|---|---|---|---|
| PGD-AT | 7.53 | 12.90 | 0.14 |
| TRADES | 6.08 | 20.70 | 0.17 |
| Ours$_{\gamma=0.0}$ | 5.47 | 33.80 | 0.21 |

Our approach follows established PAC-Bayesian frameworks (Neyshabur et al., 2017b; Farnia et al., 2019) , where the norm restriction on $\mathbf{u}$ serves as a theoretical bridge connecting worst-class error with the spectral norm of empirical confusion matrix and model weights.

The perturbation degree of $\mathbf{u}$ reflects the sharpness/flatness of the base classifier $f_\mathbf{w}$. Its effectiveness is usually evaluated in relation to generalization gaps (Jiang et al., 2020), as the single value of $\mathbf{u}$ is un-imformative. While Sec. 3 presents weight-norm-based bounds, we further validate the PAC-Bayesian bound/assumption empirically using sharpness-based sampling method (Jiang et al., 2020):

- Sample 50 perturbations $\mathbf{u}_1, ..., \mathbf{u}_{50}$ from $\mathcal{N}(0, \sigma^2\mathbf{I})$.
- Find the largest $\sigma^2 \in \{0.01, 0.02, ..., 1\}$ where training accuracy drop between $f_\mathbf{w}$ and any $f_{\mathbf{w}+\mathbf{u_i}}$ stays within 5%.

The results in Tab. 8 show that smaller generalization gaps and higher worst-class accuracy correspond to larger variances, indicating smoother base classifiers. These findings align with previous work (Jiang et al., 2020; Xiao et al., 2023), supporting the effectiveness of the PAC-Bayesian framework.

### E.4 LONG-TAIL EXPERIMENTS

Table 9: The performances on ImageNet-LT with ResNeXt-50.

| Method | Many-shot (%) | Medium-shot (%) | Few-shot (%) |
|---|---|---|---|
| Baseline | 66.1 | 38.4 | 8.9 |
| Ours$_{\gamma=0.0}$ | 61.3 | 45.5 | 31.2 |

We evaluate long-tail image classification on ImageNet-LT (Liu et al., 2019) using ResNeXt-50-32x4d as our baseline, trained with SGD (momentum 0.9, batch size 512) and cosine learning rate decay from 0.2 to 0.0 over 90 epochs. As shown in Tab. 9, our regularizer leads to improved fairness across the distribution: while showing a slight decrease in many-shot performance, it achieves significant gains in few-shot classes and moderate improvements in medium-shot classes, demonstrating effectiveness in addressing fairness problem in long-tail scenarios.

## E.5 TRAINING DYNAMIC

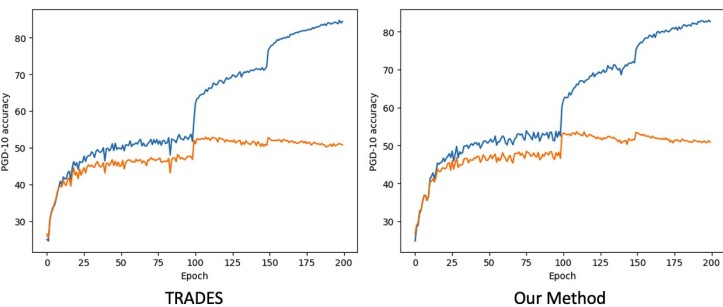

Figure 4: We adversarially trained Preact-ResNet-18 models for standard TRADES (left) and our method with $\gamma = 0.0$ based on TRADES (right) for 200 epochs using SGD with a momentum of 0.9, batch size of 256, weight decay of $5 \times 10^{-4}$, and an initial learning rate of 0.1, which is reduced by a factor of 10 at the 100th and 150th epochs. Blue line represents training accuracy under PGD-10, while orange line represents testing accuracy under PGD-10.

As illustrated in Fig. 4, both methods exhibit similar training and testing dynamics. These results suggest that our method does not independently introduce robust overfitting or address robust overfitting; rather, this phenomenon appears to be inherent to the fundamental adversarial training frameworks, e.g., TRADES, AWP, and AT.

## E.6 BEST CLASS PERFORMANCE

Table 10: Evaluation of different methods on CIFAR-10. We report the average accuracy, the worst-class accuracy, and the best-class accuracy under standard $\ell_\infty$ norm AA, with baseline models of TRADES/TRADES-AWP.

| Method | AA (%) | Worst (%) | Best (%) |
|---|---|---|---|
| TRADES | 52.51 | 23.20 | 77.10 |
| + FRL-RWRM$_{0.05}$ | 49.97 | 35.40 | 75.70 |
| + FAAL$_{AWP}$ | 52.45 | 35.40 | 77.50 |
| + Ours$_{\gamma=0.0}$ | 53.46 | 36.30 | 78.50 |
| TRADES-AWP | 56.18 | 25.80 | 77.60 |
| + FRL-RWRM$_{0.05}$ | 46.50 | 27.70 | 72.80 |
| + FAAL$_{AWP}$ | 53.93 | 37.00 | 76.20 |
| + Ours$_{\gamma=0.0}$ | 54.65 | 37.00 | 76.90 |

# F    IMPLEMENTATION DETAILS

In the current implementation version, the confusion matrix is computed over training set and the computation of gradients involves two key terms:

- The gradient of the confusion matrix's spectral norm: $\frac{\partial \|\mathcal{C}^{f\mathbf{w}}_{\mathcal{S}',\gamma}\|_2}{\partial(\mathcal{C}^{f\mathbf{w}}_{\mathcal{S}',\gamma})_{ij}}$. This is computed once per epoch over all training data using SVD, and the resulting gradient matrix is cached for use throughout the epoch.

- The final KL gradient term in (11): $\frac{\partial(\mathcal{L}^{f\mathbf{w}}_{\mathcal{S}',\gamma})_{ij}}{\partial \mathbf{w}}$, which is computed over batch set due to the computational overload of processing the entire training set at once. We believe this operation is commonly used in DNNs' training.

Then, model updates during each minibatch combine these two components.

Our method can also regularize both terms over batch set, we show empirical results in the following. We also compare our results with FAAL, as they reweight classes each minibatch. Note that in the table, Hybrid means the first spectral term in (11) is computed once per epoch over training set, and the final gradient term is comptued each minibatch over batch set. Minibatch means both terms are computed each minibatch over batch set.

Table 11:    Evaluation of regularization over epoch or minibatch on CIFAR-10. We report the training time, average accuracy and the worst-class accuracy under standard $\ell_\infty$ norm AA, with baseline models of TRADES.

| Method | AA (%) | Worst (%) | Time/Epoch (s) | Regularize on |
|--------|--------|-----------|----------------|---------------|
| TRADES | 52.51 | 23.20 | 664 | N/A |
| + FAAL$_{\text{AWP}}$ | 52.45 | 35.40 | 921 | Minibatch |
| + Ours$_{\gamma=0.0}$ | 53.46 | 36.30 | 997 | Hybrid |
| + Ours$_{\gamma=0.0}$ | 53.39 | 36.10 | 1089 | Minibatch |

To improve computational efficiency, we have implemented and evaluated an alternative version of our method. Following CFA, we utilize adversarial examples from the previous epoch to generate the confusion matrix for the current epoch.

Table 12:  Experimental setting follows the above table. Here, the confusion matrix in our method is generated by adversarial examples from previous epoch.

| Method | AA (%) | Worst (%) | Time/Epoch (s) | Regularize on |
|--------|--------|-----------|----------------|---------------|
| TRADES + Ours$_{\gamma=0.0}$ | 53.31 | 36.10 | 705 | Hybrid |

