# OpenReview forum: "Enhancing Robust Fairness via Confusional Spectral Regularization"
_ICLR.cc/2025/Conference — ICLR 2025 Poster_

### Official Review · Reviewer_ZB6w · 2024-10-30

**Soundness:** 3
**Presentation:** 4
**Contribution:** 4
**Rating:** 8
**Confidence:** 3

**Summary:**

This paper aims to enhance the worst-class accuracy under adversarial attacks. The authors first establish a robust generalization bound within the PAC-Bayesian framework, demonstrating that the worst-class error is constrained by the spectral norm of the empirical robust confusion matrix. Based on this theoretical insight, they propose a regularization loss to minimize the spectral norm of the confusion matrix. Extensive experiments on CIFAR-10, CIFAR-100, and TinyImageNet validate the effectiveness of the proposed method in improving worst-class accuracy under adversarial attacks.

**Strengths:**

(1) The paper is well-written and easy to follow. Although I am not an expert in the PAC-Bayesian framework, the authors clearly explain the flow of their proof, enabling me to understand how the main results are derived.

(2) To address the problem of non-differentiable confusion matrix, the authors propose a novel approximation method — using KL divergence to replace binary errors. This approach is conceptually sound and interesting.

(3) Improving worst-class accuracy is important but often overlooked. The authors make a meaningful contribution by providing both theoretical insights and practical methods to address this issue effectively.

**Weaknesses:**

(1) I find Paragraph 3 (Lines 51 to 66) unclear in its intended message. First, the authors state that “our findings indicate that the robust confusion matrix over the training set does not always align with that of the test set. In other words, as shown in Fig. 2.” However, Figure 2 illustrates a strong correlation between training and testing accuracy, with the worst-performing class in training being the second-worst in testing. Additionally, the performance gap between the two worst-performing classes in testing is small, which could be attributed to estimation variance. Second, the authors assert that their findings “motivate the need for a principled approach to mitigate the disparities in robust performance.” However, Proposition 3.1 suggests that the discrepancies between the training and testing confusion matrices arise from a second term, which depends on the model and training set. Minimizing this term is not a contribution of the paper and is not explicitly addressed in the work. I recommend that the authors explain more explicitly how their findings motivate their approach.

(2) Lines 182–184: It is not immediately clear why the worst-class robust error can be represented as the L_1 norm of the confusion matrix. Could the authors provide a detailed derivation to clarify this relationship?

(3) There is an inconsistency between the confusion matrix in Figure 3 and the one in Eq. (1). In Eq. (1), the diagonal entries are defined as 0, whereas in Figure 3, the diagonal contains non-zero values. I suggest that the authors clarify this discrepancy and modify either Figure 3 or Equation 1 to be consistent, if appropriate.


(4) The experiments in this paper focus solely on adversarial training and evaluate both clean and robust worst-class accuracy. According to Lemma 3.3, the proposed method is also applicable to improving worst-class accuracy under clean training (e.g., using standard cross-entropy loss). I am curious whether the proposed method can enhance worst-class accuracy under clean training setting (CIFAR100 and ImageNet results), as demonstrated in [1]. For example, conduct additional experiments on CIFAR100 and ImageNet using standard cross-entropy loss.

(5) Why is the sign() function required in Eq. (11)? Why can’t the average KL divergence be used directly? Are there any ablation studies to justify this operation? I suggest that authors include ablation studies comparing performance with and without the sign() function or
discuss any potential drawbacks or limitations of using the average KL divergence directly.

minor typos: Line 470, Tab. 2 should be Tab. 3.

[1] Classes Are Not Equal: An Empirical Study on Image Recognition Fairness

**Questions:**

See weaknesses. I am open to increasing my score if the identified weaknesses are adequately addressed.

---

> ### Author Response · Authors · 2024-11-23
> **Response by authors 1/4**
>
> Dear Reviewer ZB6w,
>
> We sincerely appreciate your comments about motivation, typos, experiments, and Eq. (11).
>
> We have revised the typos in the updated manuscript, in the following, we address other concerns one-by-one.
>
> ---
>
> **Q1(1):  I find Paragraph 3 (Lines 51 to 66) unclear in its intended message. First, the authors state that “our findings indicate that the robust confusion matrix over the training set does not always align with that of the test set. In other words, as shown in Fig. 2.” However, Figure 2 illustrates a strong correlation between training and testing accuracy, with the worst-performing class in training being the second-worst in testing. Additionally, the performance gap between the two worst-performing classes in testing is small, which could be attributed to estimation variance. ..., I recommend that the authors explain more explicitly how their findings motivate their approach.**
>
> **A1(1):** We apologize for any confusion and have revised the third paragraph to better articulate the connection between our motivation and method. Our key observation is that existing robust fairness approaches (e.g., FRL and FAAL) explicitly reweight classes based on training set performance during adversarial training. However, we find that different classes exhibit varying generalization gaps between training and test set performance. This class-dependent divergence means that training set performance may not reliably predict test set performance, potentially limiting the effectiveness of previous explicit reweighting methods. Moreover, as demonstrated in revised **Fig. 2 (right)**, these methods can actually amplify the training-test divergence **(lower training-test correlation)**, consequently hindering their ability to optimize worst-class robust accuracy.
>
> Our approach takes a fundamentally different path: instead of explicitly reweighting classes based on empirical class performance, we regularize the spectral norm of the empirical confusion matrix to improve worst-class robust performance. This approach is theoretically grounded in PAC-Bayesian bounds and, empirically, achieves both lower training-test divergence (higher training-test correlation, see Fig. 2) and better worst-class robust performance compared to FRL and FAAL.
>
> In conclusion, while training-test divergence can limit the effectiveness of explicit class-reweighting methods (and may be exacerbated by them), our theoretically-motivated spectral norm regularization provides an alternative that avoids these pitfalls. Our empirical results demonstrate that this approach not only maintains better alignment between training and test performance but also achieves superior worst-class robust accuracy. **Note that our method is not primarily designed to address the training-test divergence, which may be inherent to the dataset, but rather to avoid explicit reweighting that can exacerbate this divergence and limit effectiveness.**
>
> **Justification:** (1) The empirical results in revised Fig. 2
> (2) The explicit reweighting strategy proposed in FRL, FAAL, ...
> (3) The spectral regularizer (distinguish from explicit reweighting) shown in this manuscript
> (4) The experiments shown in this manuscript
>
> ---
>
> **Q1(2): Second, the authors assert that their findings “motivate the need for a principled approach to mitigate the disparities in robust performance.” However, Proposition 3.1 suggests that the discrepancies between the training and testing confusion matrices arise from a second term, which depends on the model and training set. Minimizing this term is not a contribution of the paper and is not explicitly addressed in the work.**
>
> **A1(2):** Prop. 3.1 establishes a robust generalization bound for worst-class performance, which depends on two components: the empirical spectral norm and a model/training data-dependent term. The second term suggests three approaches to tighten the bound:
>
> * Increasing the minimum number of training examples ($m_{min}$)
> * Regularizing the Frobenius norm of weights ($||{\bf W}||_F$)
> * Regularizing the spectral norm of weights ($||{\bf W}||_2$)
>
> While the first two approaches are addressed by common techniques (e.g., data augmentation, weight decay, batch normalization), and the third has been explored but is computationally expensive (Farnia et al. 2019), our key theoretical contribution is identifying a new, efficiently optimizable term: the spectral norm of the confusion matrix. This discovery theoretically motivates our proposed regularization approach for improving worst-class performance.
>
> [1] Farzan Farnia, Jesse M Zhang, and David Tse. Generalizable adversarial training via spectral normalization. In ICLR, 2019.
>
> ---

---

> ### Author Response · Authors · 2024-11-23
> **Response by authors 2/4**
>
> **Q2: Lines 182–184: It is not immediately clear why the worst-class robust error can be represented as the L_1 norm of the confusion matrix. Could the authors provide a detailed derivation to clarify this relationship?**
>
> **A2:** Look at the definition in (1), one could find the sum of the $j$-th column of ${\mathcal{C}}^{f_ {\bf w}} _{\mathcal{D}'}$ represents the expected error for class j, which can be expressed as:
>
> $\sum_i ({\mathcal{C}}^{f_ {\bf w}} _{\mathcal{D}'}) _{ij}= \underset{(\mathbf{x}', y) \sim \mathcal{D}'}{\mathbb{P}} (\max _{i\ne j} f _{\bf w}({\bf x}')[i] \ge f _{\bf w}({\bf x}')[j] \mid y=j).$
>
> Since the $\ell _1$ matrix norm represents the maximum column sum of absolute elements, it naturally corresponds to the worst-class error, allowing us to express it as
>
> $||{\mathcal{C}}^{f_{\bf w}} _{\mathcal{D}'}||_1 =\max _j \sum_i ({\mathcal{C}}^{f _ {\bf w}} _{\mathcal{D}'}) _{ij}.$
>
> We have added this derivation to the revised manuscript for clarity.
>
> ---
>
> **Q3: There is an inconsistency between the confusion matrix in Figure 3 and the one in Eq. (1). In Eq. (1), the diagonal entries are defined as 0, whereas in Figure 3, the diagonal contains non-zero values. I suggest that the authors clarify this discrepancy and modify either Figure 3 or Equation 1 to be consistent, if appropriate.**
>
> **A3:** Thank the reviewer for noting this discrepancy. We acknowledge that Fig. 3's visualization was inconsistent with the formal definition in (1), as we discussed in the original manuscript. While we initially included diagonal values for visual completeness, we have now revised Fig. 3 to strictly follow the definition in (1), where diagonal entries are zero. This revision ensures consistency between our definition and visual representation.
>
> ---
>
> **Q4: The experiments in this paper focus solely on adversarial training and evaluate both clean and robust worst-class accuracy. According to Lemma 3.3, the proposed method is also applicable to improving worst-class accuracy under clean training (e.g., using standard cross-entropy loss). I am curious whether the proposed method can enhance worst-class accuracy under clean training setting (CIFAR100 and ImageNet results), as demonstrated in [1]. For example, conduct additional experiments on CIFAR100 and ImageNet using standard cross-entropy loss.**
>
>
> **A4:** We appreciate the reviewer's suggestion for broader empirical validation. Following Cui et al. (2024), we have extended our evaluation to include experiments on standard ImageNet and CIFAR-100 in clean training settings (detailed in **App. E.2**). For your convenience, we present these new results here.
>
> | **Dataset**   |            | **Method**                 |            | **Easy (%)**      | **Medium (%)**    | **Hard (%)**        | **All (%)**      |
> |---------------|------------|----------------------------|------------|--------------------|--------------------|---------------------|------------------|
> | **ImageNet**  |            | Baseline (ResNet-50)      |            | 93.1              | 81.1              | 59.4               | 77.8            |
> |               |            | Menon et al. (2021)      |            | 91.7 (-1.4)       | 79.8 (-1.3)       | 61.4 (+2.0)        | 77.6            |
> |               |            | Cui et al. (2019)       |            | 91.6 (-1.5)       | 79.6 (-1.5)       | 61.3 (+1.9)        | 77.5            |
> |               |            | **Ours$_{\gamma=0.0}$**        |            | 91.5 (-1.6)   | 79.6 (-1.5)   | **62.3 (+2.9)**    | **77.8**        |
> | **CIFAR-100** |            | Baseline (WRN-34-10)      |            | 92.2              | 83.2              | 70.1               | 81.7            |
> |               |            | Menon et al. (2021)      |            | 91.5 (-0.7)       | 83.2 (+0.0)       | 70.4 (+0.3)        | 81.6            |
> |               |            | Nam et al. (2020)        |            | 90.4 (-1.8)       | 81.5 (-1.7)       | 67.7 (-2.4)        | 79.7            |
> |               |            | Liu et al., (2021)        |            | 91.3 (-0.9)       | 82.4 (-0.8)       | 69.6 (-0.5)        | 81.0            |
> |               |            | **Ours$_{\gamma=0.0}$**        |            | 91.3 (-0.9)   | 83.0 (-0.2)  | **71.2 (+1.1)**    | **81.8**        |
>
>
> The expanded experiments demonstrate that our method's effectiveness generalizes to larger-scale datasets: it successfully enhances fairness in clean training by improving test accuracy on "hard" subsets of both ImageNet and CIFAR-100 while maintaining overall accuracy. These results provide additional evidence for the broader applicability of our approach across different dataset scales and complexity levels.

---

> ### Author Response · Authors · 2024-11-23
> **Responses by authors 3/4**
>
> | **Method**          |            | **Many-Shot (%)** |            | **Medium-Shot (%)** |            | **Few-Shot (%)** |
> |----------------------|------------|-------------------|------------|---------------------|------------|------------------|
> | **Baseline**        |            | 66.1             |            | 38.4               |            | 8.9              |
> | **Ours$_{\gamma=0.0}$**  |            | 61.3         |            | 45.5           |            | 31.2         |
>
>
> In addition, we evaluate long-tail image classification on ImageNet-LT (Liu et al., 2019) using ResNeXt-50-32x4d as our baseline, trained with SGD (momentum 0.9, batch size 512) and cosine learning rate decay from 0.2 to 0.0 over 90 epochs.
> As shown in the table, our regularizer leads to improved fairness across the distribution: while showing a slight decrease in many-shot performance, it achieves significant gains in few-shot classes and moderate improvements in medium-shot classes, demonstrating effectiveness in addressing fairness problem in long-tail scenarios.
>
> [1] Jiequan Cui, Beier Zhu, Xin Wen, Xiaojuan Qi, Bei Yu, and Hanwang Zhang. Classes are not equal: An empirical study on image recognition fairness. In Proceedings of the IEEE/CVF Conference on Computer Vision and Pattern Recognition, pp. 23283–23292, 2024.
> [2] Aditya Krishna Menon, Sadeep Jayasumana, Ankit Singh Rawat, Himanshu Jain, Andreas Veit, and Sanjiv Kumar. Long-tail learning via logit adjustment. In International Conference on Learning Representations, 2021.
> [3] Yin Cui, Menglin Jia, Tsung-Yi Lin, Yang Song, and Serge Belongie. Class-balanced loss based on effective number of samples. In Proceedings of the IEEE/CVF conference on computer vision and pattern recognition, pp. 9268–9277, 2019.
> [4] Junhyun Nam, Hyuntak Cha, Sungsoo Ahn, Jaeho Lee, and Jinwoo Shin. Learning from failure: De-biasing classifier from biased classifier. Advances in Neural Information Processing Systems, 33:20673–20684, 2020.
> [5] Evan Z Liu, Behzad Haghgoo, Annie S Chen, Aditi Raghunathan, Pang Wei Koh, Shiori Sagawa, Percy Liang, and Chelsea Finn. Just train twice: Improving group robustness without training group information. In International Conference on Machine Learning, pp. 6781–6792. PMLR, 2021.
> [6] Ziwei Liu, Zhongqi Miao, Xiaohang Zhan, Jiayun Wang, Boqing Gong, and Stella X Yu. Large scale long-tailed recognition in an open world. In Proceedings of the IEEE/CVF conference on computer vision and pattern recognition, pp. 2537–2546, 2019.
>
> ---
>
> **Q5: Why is the sign() function required in Eq. (11)? Why can’t the average KL divergence be used directly? Are there any ablation studies to justify this operation? I suggest that authors include ablation studies comparing performance with and without the sign() function or discuss any potential drawbacks or limitations of using the average KL divergence directly.**
>
> **A5:** We appreciate the reviewer's comment and have revised our presentation in Sec. 4.1. Specifically, we have rewritten (9) and (11) to use $\sum$ notation for unit gradient summation rather than the potentially misleading $\frac{vector}{vector}$ notation.
>
> Revised (9):
> $$
> \underbrace{\frac{\partial || {\mathcal{C}}^{f _{\bf w}} _{{\mathcal{S}}',\gamma}|| _2}{\partial {\bf w}}} _{\textbf{Expensive}} \Longrightarrow
> \sum _{i\ne j}\underbrace{\frac{\partial || {\mathcal{C}}^{f _{\bf w}} _{{\mathcal{S}}',\gamma}|| _2}{\partial  ({\mathcal{C}}^{f _{\bf w}} _{{\mathcal{S}}',\gamma}) _{ij}}} _{\textbf{Cheap}} \times \underbrace{\frac{\partial ({\mathcal{C}}^{f _{\bf w}} _{{\mathcal{S}}',\gamma}) _{ij}}{\partial{\bf w}}} _{\textbf{Expensive}} \Longrightarrow
> \sum _{i\ne j}\underbrace{\frac{\partial || {\mathcal{C}}^{f _{\bf w}} _{{\mathcal{S}}',\gamma}|| _2}{\partial ({\mathcal{C}}^{f _{\bf w}} _{{\mathcal{S}}',\gamma}) _{ij}}} _{\textbf{Cheap}} \times \underbrace{\frac{\partial ({\mathcal{C}}^{f _{\bf w}} _{{\mathcal{S}}',\gamma}) _{ij}}{\partial ({\mathcal{L}}^{f _{\bf w}} _{{\mathcal{S}}',\gamma}) _{ij}}} _{\textbf{Approximate}} \times \underbrace{\frac{\partial ({\mathcal{L}}^{f _{\bf w}} _{{\mathcal{S}}',\gamma}) _{ij}}{\partial {\bf w}}} _{\textbf{Cheap}}
> $$
>
>
> Revised (11):
> $$
> \frac{\partial \Psi(f _{\bf w},\mathcal{S}',\gamma)}{\partial {\bf w}} = \sum _{i\ne j} \Bigg [ \frac{\partial || \mathcal{C}^{f _{\bf w}} _{\mathcal{S}',\gamma}|| _2}{\partial (\mathcal{C}^{f _{\bf w}} _{\mathcal{S}',\gamma}) _{ij}} \times \text{sign}\Bigg( \frac{\partial (\mathcal{C}^{f _{\bf w}} _{\mathcal{S}',\gamma}) _{ij}}{\partial (\mathcal{L}^{f _{\bf w}} _{\mathcal{S}',\gamma}) _{ij}}\Bigg) \times \frac{\partial (\mathcal{L}^{f _{\bf w}} _{\mathcal{S}',\gamma}) _{ij}}{\partial {\bf w}} \Bigg ]
> $$

---

> ### Author Response · Authors · 2024-11-23
> **Response By authors 4/4**
>
> In the right-hand side of (9), we approximate the general direction (sign) of $\partial (\mathcal{C}^{f _{\bf w}} _{\mathcal{S}',\gamma}) _{ij}/\partial ({\mathcal{L}}^{f _{\bf w}} _{\mathcal{S}',\gamma}) _{ij}$ based on the observation that the descent direction of non-diagonal elements in $\mathcal{C}^{f _{\bf w}} _{\mathcal{S}',\gamma}$ closely aligns with that of ${\mathcal{L}}^{f{\bf w}} _{\mathcal{S}',\gamma}$. This leads to our approximation: $$\text{sign}\Bigg( \frac{\partial (\mathcal{C}^{f _{\bf w}} _{\mathcal{S}',\gamma}) _{ij}}{\partial (\mathcal{L}^{f _{\bf w}} _{\mathcal{S}',\gamma}) _{ij}}\Bigg)\approx 1.$$
> This approximation strategy parallels common practice in machine learning. For example, in classification tasks, we cannot directly compute gradients with respect to error (accuracy), so we optimize differentiable proxies like cross-entropy loss or KL divergence, assuming their optimization directions align. Similarly, we adopt this approximation for **computational efficiency**, as accurately computing the exact gradient term would be prohibitively complex.
>
> **In conclusion**, the gradient inside the sign function cannot be used directly due to the non-differentiable nature of 0-1 errors in $\mathcal{C}^{f _{\bf w}} _{\mathcal{S}',\gamma}$ (high complexity to estimate its discrete gradient). Therefore, we approximate the gradient direction using the sign() function, effectively treating it as 1. We would like to point out that this approximation strategy is commonly used in machine learning, e.g., optimize differentiable loss functions (e.g., cross-entropy, KL divergence) to reduce non-differentiable error.
>
> ---
>
> We appreciate your detailed comments and hope we have addressed your concerns. Please let us know if you have any additional questions.
>
> Sincerely,
> Authors

---

> > ### Comment · Reviewer_ZB6w · 2024-11-24
> >
> > Thank you for the authors’ responses, which have addressed my concerns. I have raised my score.
> > Good luck with your submission!

---

> ### Author Response · Authors · 2024-11-24
> **Thank the Reviewer ZB6w**
>
> We are delighted to see our score has been improved to 8. Your thoughtful suggestions have significantly enhanced the quality of our manuscript. Thank you for your time and dedication throughout this review process.

---

### Official Review · Reviewer_RdH6 · 2024-11-01

**Soundness:** 3
**Presentation:** 4
**Contribution:** 3
**Rating:** 6
**Confidence:** 4

**Summary:**

This paper addresses the issue of fairness in adversarial robustness. The authors leverage PAC-based bounds to analyze the worst-class error. They first provide a bound for deterministic classifiers based on the PAC framework. Next, using techniques proposed by Neyshabur et al., they bound the KL divergence, which is challenging to compute. Finally, they apply the analysis from Xiao et al. to obtain Lemma 3.4. Empirically, based on their analysis of the confusion matrix, they propose the addition of an approximated regularization term. Empirical results support the effectiveness of the proposed method.

**Strengths:**

This paper is well-motivated, clearly written, and easy to follow.

The theoretical results are novel and offer an interesting perspective on the problem of robustness fairness.

Empirically, the proposed method enhances worst-class performance.

**Weaknesses:**

Concerns regarding this paper primarily focus on practical implications. From a methodological perspective, the authors apply a regularization term to the confusion matrix. Intuitively, calculating the KL divergence between misclassified examples is akin to re-examining those examples or, in other words, assigning greater weight to misclassified instances. Since only the empirical confusion matrix is used for optimization during training, this method may not effectively address their motivation, which is to resolve the mismatch between training and testing worst-class performance.

As a result, the proposed method may encounter issues such as sacrificing the performance of the best class and an increased likelihood of robust overfitting, both of which are evident in the empirical results presented in this paper.

Regarding the experimental setup, the authors overlook important evaluations under PGD attacks. Each experimental condition should be assessed across all three datasets rather than just one or two, and both $L_2$ and $L_\infty$ attacks should be included.

**Questions:**

Please see weaknesses.

---

> ### Author Response · Authors · 2024-11-22
> **Response by authors 1/4**
>
> Dear Reviewer RdH6,
>
> We sincerely appreciate your comments about motivation, trade-off, and more experiments.
>
> In the following, we address the raised concerns one-by-one.
>
> ---
>
> **Q1(1): Concerns regarding this paper primarily focus on practical implications. From a methodological perspective, the authors apply a regularization term to the confusion matrix. Intuitively, calculating the KL divergence between misclassified examples is akin to re-examining those examples or, in other words, assigning greater weight to misclassified instances.**
>
> **A1(1):** We appreciate the reviewer's comment, **we agree that our method regularizes the confusion matrix (errors from misclassified examples) but would like to emphasize that our method differs fundamentally from previous class-reweighting approaches.** As detailed in **revised Sec. 4.1**, one of our key contributions is the regularization of the confusion matrix's spectral norm. The technical challenge lies in optimizing this spectral norm, as the discrete nature of confusion matrix elements makes them non-differentiable and incompatible with gradient-based optimization. To address this, (9) demonstrates our solution: approximating the non-differentiable binary error terms using KL divergence to make the objective differentiable.
>
> This approximation strategy is well-established in the machine learning community, similar to how classification tasks typically optimize differentiable loss functions (e.g., cross-entropy, KL divergence) rather than directly optimizing accuracy (error). Thus, rather than assigning greater weight to misclassified instances, our use of KL divergence serves a fundamentally different purpose: enabling gradient-based optimization of our theoretically-motivated spectral norm regularizer. **Importantly, since our method updates the model based on the spectral norm of the confusion matrix, worse-performing classes on training set do not automatically receive greater weight during training. Technically, this is the main difference between ours and previous reweighting methods.**
>
> ---
>
> **Q1(2): Since only the empirical confusion matrix is used for optimization during training, this method may not effectively address their motivation, which is to resolve the mismatch between training and testing worst-class performance.**
>
> **A1(2):** We apologize for any confusion and have revised the third paragraph to better articulate the connection between our motivation and method. Our key observation is that existing robust fairness approaches (e.g., FRL and FAAL) explicitly reweight classes based on training set performance during adversarial training. However, we find that different classes exhibit varying generalization gaps between training and testing set performance. This class-dependent divergence means that training set performance may not reliably predict test set performance, potentially limiting the effectiveness of previous explicit reweighting methods. Moreover, as demonstrated in revised **Fig. 2 (right)**, these methods can actually amplify the training-test divergence **(lower training-test correlation)**, consequently hindering their ability to optimize worst-class robust accuracy.
>
> Our approach takes a fundamentally different path: instead of explicitly reweighting classes based on empirical class performance, we regularize the spectral norm of the empirical confusion matrix to improve worst-class robust performance. This approach is theoretically grounded in PAC-Bayesian bounds and, empirically, achieves both lower training-test divergence (higher training-test correlation, see Fig. 2) and better worst-class robust performance compared to FRL and FAAL.
>
> In conclusion, while training-test divergence can limit the effectiveness of explicit class-reweighting methods (and may be exacerbated by them), our theoretically-motivated spectral norm regularization provides an alternative that avoids these pitfalls. Our empirical results demonstrate that this approach not only maintains better alignment between training and test performance but also achieves superior worst-class robust accuracy. **Note that our method is not primarily designed to address the training-test divergence, which may be inherent to the dataset, but rather to avoid explicit reweighting that can exacerbate this divergence and limit effectiveness to improve worst-class performance.**
>
> **Justification:** (1) The empirical results in revised Fig. 2;
> (2) The explicit reweighting strategy proposed in FRL, FAAL, ...
> (3) The spectral regularizer (distinguish from explicit reweighting) shown in this manuscript
> (4) The experiments shown in this manuscript
>
> ---

---

> ### Author Response · Authors · 2024-11-22
> **Response by authors 2/4**
>
> **Q2: As a result, the proposed method may encounter issues such as sacrificing the performance of the best class and an increased likelihood of robust overfitting, both of which are evident in the empirical results presented in this paper.**
>
> **A2:**
> * As shown in Tab. 1, for TRADES-trained models, our method achieves both the best average AA accuracy and the best worst-class AA accuracy, demonstrating no performance sacrifice. For TRADES-AWP-trained models, while there is a slight trade-off in average AA accuracy, our method still achieves the best worst-class AA accuracy with a smaller decrease in average performance compared to other robust fairness fine-tuning methods. Notably, our approach outperforms both FAAL and FRL - the current state-of-the-art methods in this fine-tuning scenario - in terms of both average and worst-class AA accuracy.
> * As demonstrated in Tab. 2, our method can actually improve average AA accuracy in some cases - for example, with CIFAR-10 under $\ell_2$ attack using WRN-70-16 architecture, we observe an increase from 84.86% to 85.06% in average AA accuracy. In other cases, while there may be a minor trade-off in average AA accuracy, this is compensated by significant improvements in worst-class AA accuracy.
> * In Tab. 3, when training from scratch, our method improves both worst-class and average AA accuracies compared to standard adversarial training methods (PGD-AT and TRADES). Moreover, it outperforms other robust fairness methods in most cases, with the average AA accuracy on CIFAR-10 falling only marginally (0.18%) behind CFA while achieving better worst-class performance.
>
> Our empirical results demonstrate that improving worst-class robustness does not necessarily come at the cost of overall performance. In fine-tuning scenarios, while our method may introduce a modest trade-off in average AA accuracy, it achieves significantly better worst-class robust accuracy and outperforms  FAAL and FRL. More notably, when training from scratch, our approach improves both average and worst-class AA accuracies compared to standard adversarial training methods (PGD-AT and TRADES) and consistently outperforms other robust fairness methods. These comprehensive results across different training scenarios validate the effectiveness of our approach, as it demonstrates superior performance in both worst-class and average accuracy metrics **compared to existing robust fairness methods.**
>
> Following Cui et al. (2024), we have extended our evaluation to include experiments on standard ImageNet and CIFAR-100 in clean training settings (detailed in Appendix E.2). For the your convenience, we present these new results here.
>
> | **Dataset**   |            | **Method**                 |            | **Easy (%)**      | **Medium (%)**    | **Hard (%)**        | **All (%)**      |
> |---------------|------------|----------------------------|------------|--------------------|--------------------|---------------------|------------------|
> | **ImageNet**  |            | Baseline (ResNet-50)      |            | 93.1              | 81.1              | 59.4               | 77.8            |
> |               |            | Menon et al. (2021)      |            | 91.7 (-1.4)       | 79.8 (-1.3)       | 61.4 (+2.0)        | 77.6            |
> |               |            | Cui et al. (2019)       |            | 91.6 (-1.5)       | 79.6 (-1.5)       | 61.3 (+1.9)        | 77.5            |
> |               |    | **Ours$_{\gamma=0.0}$**        |            | 91.5 (-1.6)   | 79.6 (-1.5)   | **62.3 (+2.9)**    | **77.8**        |
> | **CIFAR-100** |            | Baseline (WRN-34-10)      |            | 92.2              | 83.2              | 70.1               | 81.7            |
> | |   | Menon et al. (2021)      |            | 91.5 (-0.7)       | 83.2 (+0.0)       | 70.4 (+0.3)        | 81.6            |
> |     |  | Nam et al. (2020)        |            | 90.4 (-1.8)       | 81.5 (-1.7)       | 67.7 (-2.4)        | 79.7            |
> |   |    | Liu et al., (2021)        |            | 91.3 (-0.9)       | 82.4 (-0.8)       | 69.6 (-0.5)        | 81.0            |
> |   |            | **Ours$_{\gamma=0.0}$**        |            | 91.3 (-0.9)   | 83.0 (-0.2)   | **71.2 (+1.1)**    | **81.8**        |
>
> The expanded experiments demonstrate that our method's effectiveness generalizes to larger-scale datasets: it successfully enhances fairness in clean training by improving test accuracy on "hard" subsets of both ImageNet and CIFAR-100 while maintaining overall accuracy. These results provide additional evidence for the broader applicability of our approach across different dataset scales and complexity levels.

---

> ### Author Response · Authors · 2024-11-22
> **Response by authors 3/4**
>
> | **Method**          |            | **Many-Shot (%)** |            | **Medium-Shot (%)** |            | **Few-Shot (%)** |
> |----------------------|------------|-------------------|------------|---------------------|------------|------------------|
> | **Baseline**        |            | 66.1             |            | 38.4               |            | 8.9              |
> | **Ours$_{\gamma=0.0}$**  |            | 61.3         |            | 45.5           |            | 31.2         |
>
>
> In addition, we evaluate long-tail image classification on ImageNet-LT (Liu et al., 2019) using ResNeXt-50-32x4d as our baseline, trained with SGD (momentum 0.9, batch size 512) and cosine learning rate decay from 0.2 to 0.0 over 90 epochs.
> As shown in the table, our regularizer leads to improved fairness across the distribution: while showing a slight decrease in many-shot performance, it achieves significant gains in few-shot classes and moderate improvements in medium-shot classes, demonstrating effectiveness in addressing fairness problem in long-tail scenarios.
>
> [1] Jiequan Cui, Beier Zhu, Xin Wen, Xiaojuan Qi, Bei Yu, and Hanwang Zhang. Classes are not equal: An empirical study on image recognition fairness. In Proceedings of the IEEE/CVF Conference on Computer Vision and Pattern Recognition, pp. 23283–23292, 2024.
> [2] Aditya Krishna Menon, Sadeep Jayasumana, Ankit Singh Rawat, Himanshu Jain, Andreas Veit, and Sanjiv Kumar. Long-tail learning via logit adjustment. In International Conference on Learning Representations, 2021.
> [3] Yin Cui, Menglin Jia, Tsung-Yi Lin, Yang Song, and Serge Belongie. Class-balanced loss based on effective number of samples. In Proceedings of the IEEE/CVF conference on computer vision and pattern recognition, pp. 9268–9277, 2019.
> [4] Junhyun Nam, Hyuntak Cha, Sungsoo Ahn, Jaeho Lee, and Jinwoo Shin. Learning from failure: De-biasing classifier from biased classifier. Advances in Neural Information Processing Systems, 33:20673–20684, 2020.
> [5] Evan Z Liu, Behzad Haghgoo, Annie S Chen, Aditi Raghunathan, Pang Wei Koh, Shiori Sagawa, Percy Liang, and Chelsea Finn. Just train twice: Improving group robustness without training group information. In International Conference on Machine Learning, pp. 6781–6792. PMLR, 2021.
> [6] Ziwei Liu, Zhongqi Miao, Xiaohang Zhan, Jiayun Wang, Boqing Gong, and Stella X Yu. Large scale long-tailed recognition in an open world. In Proceedings of the IEEE/CVF conference on computer vision and pattern recognition, pp. 2537–2546, 2019.
>
> ---
>
> **Q3: Regarding the experimental setup, the authors overlook important evaluations under PGD attacks. Each experimental condition should be assessed across all three datasets rather than just one or two, and both $\ell_2$ and $\ell_\infty$ attacks should be included.**
>
> **A3:** We initially focused our evaluations on AutoAttack (AA) as it represents one of the strongest attack methods and is widely accepted as a comprehensive benchmark for adversarial robustness. However, we appreciate the reviewer's suggestion for more extensive evaluations. We have now expanded our experiments to include PGD/CW attacks and $\ell_2$ perturbations across our datasets, with results presented in Tabs. 4 and 5. These new results consistently demonstrate that our method effectively improves worst-class robust accuracy while maintaining competitive average robust accuracy across different attack types and datasets.
>
> While we were unable to replicate some experimental conditions from prior works during the rebuttal period (we have tried our best to conduct more experiments), our expanded evaluation provides strong evidence for the generalizability of our approach across different attack types and datasets. For your convenience, we move the tables here.
>
> 1. We provide more results in revised Tab. 4, with evaluation of different fine-tuning methods on CIFAR-10 with WRN-34-10. We report the average accuracy and the worst-class accuracy under standard $\ell_\infty$ norm PGD-20/CW-20 attack, with baseline models of TRADES and TRADES-AWP. For your convenience, we move the table here.

---

> ### Author Response · Authors · 2024-11-23
> **Response by authors 4/4**
>
> | **Fine-Tuning Method**    |            | **PGD-20 (%)** | **Worst (%)** | **CW-20 (%)** | **Worst (%)** |            | **PGD-20 (%)** | **Worst (%)** | **CW-20 (%)** | **Worst (%)** |
> |---------------------------|------------|----------------|---------------|----------------|---------------|------------|----------------|---------------|----------------|---------------|
> |                           |            | **TRADES**     |               |                |               |            | **TRADES-AWP** |               |                |               |
> | **TRADES/AWP**            |            | 55.32          | 27.10         | 53.92          | 24.80         |            | **59.20**      | 28.80         | **57.14**      | 26.50         |
> | + FRL-RWRM$_{0.05}$       | | 53.16          | 40.60         | 51.39          | 36.30         |  | 49.90          | 31.70         | 49.68          | 34.00         |
> | + FRL-RWRM$_{0.07}$       | | 53.76          | 39.20         | 52.92          | 36.80         |            | 48.63          | 30.90         | 49.77          | 31.50         |
> | + FAAL$_{\text{AT}}$      |    | 53.46          | 39.80         | 52.72          | 38.20         |  | 52.54          | 35.00         | 51.70          | 34.40         |
> | + FAAL$_{\text{AWP}}$     |   | 56.07          | 43.30         | 54.16          | 38.60         |  | 57.14          | 43.40         | 55.34          | 40.10         |
> | **+ Ours$_{\gamma=0.0}$** |   | **57.83**      | **44.60**     | **56.09**      | **39.70**     |   | 59.06          | 44.10         | 56.79          | 41.30         |
> | **+ Ours$_{\gamma=0.1}$** |  | 57.66          | 44.10         | 55.97          | 38.90         |  | 58.87          | **44.50**     | 56.44          | **41.80**     |
>
> 2. We provide more results in revised Tab. 5, with evaluation of  different methods training from scratch on CIFAR-10 using Preact-ResNet18 model with standard $\ell_2$ norm attack.
>
> | **Method**    |    | **Clean (%)** | **Worst (%)** |   | **PGD-20 (%)** | **Worst (%)** |  | **CW-20 (%)** | **Worst (%)** |            | **AA (%)** | **Worst (%)** |
> |------------------------|------------|---------------|---------------|------------|----------------|---------------|------------|----------------|---------------|------------|------------|---------------|
> | **PGD-AT**            |  | 88.83         | 74.90         |   | 68.83          | 43.50         |    | 68.61          | 43.20         | | 67.99      | 42.60   |
> | **FAAL$_{\text{AT}}$** |    | 87.61         | 76.30         |   | 66.57          | 46.80         |   | 66.31          | 46.80         |    | 65.45      | 44.20   |
> | **FAAL$_{\text{TRADES}}$** |        | 86.62         | 78.10         | | 65.21          | 48.20         |   | 65.04          | 48.10         |   | 64.25      | 46.40         |
> | **Ours$_{\gamma=0.0}$** |  | **89.57**     | **78.80**     |  | **70.36**      | 49.10         |   | **69.98**      | 48.90   |  | **69.51**  | 47.50         |
> | **Ours$_{\gamma=0.1}$** |   | 89.36         | 78.70         |  | 70.16          | **49.90**     |  | 69.74          | **49.60**     |  | 69.32      | **48.40**     |
>
> ---
>
> We appreciate your insightful comments and hope we have addressed your concerns. Please let us know if you have any additional questions.
>
> Sincerely,
> Authors

---

> > ### Comment · Reviewer_RdH6 · 2024-11-25
> >
> > I appreciate the authors' rebuttal, which generally addresses my concerns. The explanations regarding motivation are clearer, and the additional experiments on PGD and CW effectively support the proposed method.
> >
> > If I understand correctly, as referenced in the supplementary code, instead of directly reweighting based on the statistical confusion matrix, the proposed method leverages SVD and only keeps the information related to the largest singular value. Since the motivation states that the training confusion matrix may be a biased estimation of the testing confusion matrix, introducing such a process as a denoising technique seems reasonable. However, it is still not very clear to me how this modification (introducing SVD) is supported by the experiments. The concern becomes more serious with the following issue:
> >
> > While reviewing the code, another question arose: the confusion matrix is calculated on the full dataset instead of a batch of data, as shown in Lines 125–146 in train_fair.py. This seems inconsistent with existing methods (which calculate the confusion matrix within a batch) and introduces extra training cost (about double the training time, as the adversarial examples are calculated twice). Calculating the confusion matrix from the full data decreases its bias with respect to the testing confusion matrix. It is now hard to distinguish the contributions between SVD and the method of confusion matrix calculation.
> >
> > Furthermore, my second question—'As a result, the proposed method may encounter issues such as sacrificing the performance of the best class and an increased likelihood of robust overfitting, both of which are evident in the empirical results presented in this paper'—is not well addressed. My primary concern is about the performance of the best class, rather than the average performance. Regarding the robust overfitting issue, it would be convincing if a training dynamics graph is provided.
> >
> > Although I recognize the progress made in the manuscript during the rebuttal, these current issues are still crucial, and I cannot change my rating to acceptance at this time.

---

> ### Author Response · Authors · 2024-11-26
> **Response by Authors 1/3**
>
> Thank you for your thorough review and thoughtful feedback on our manuscript.
>
> We are grateful to hear that the manuscript's clarity has improved and sincerely appreciate your recognition of its strengths.
>
> In the following, we address your further concerns one-by-one.
>
> ---
>
> **Q4: If I understand correctly, as referenced in the supplementary code, instead of directly reweighting based on the statistical confusion matrix, the proposed method leverages SVD and only keeps the information related to the largest singular value. Since the motivation states that the training confusion matrix may be a biased estimation of the testing confusion matrix, introducing such a process as a denoising technique seems reasonable. However, it is still not very clear to me how this modification (introducing SVD) is supported by the experiments.**
>
> **A4**: Sorry for any misunderstanding about the role of SVD in our implementation. To clarify, **SVD is used specifically for computing the gradients of the confusion matrix's spectral norm**, i.e., the first term in (11):
> $$\frac{\partial || {\mathcal{C}}^{f _{\bf w}} _{{\mathcal{S}}',\gamma}|| _2}{\partial  ({\mathcal{C}}^{f _{\bf w}} _{{\mathcal{S}}',\gamma}) _{ij}}.$$
>
> This computation relies on a fundamental relationship between a matrix's spectral norm and its singular value decomposition. For a matrix $\bf A$ with SVD:
> $$\bf A = U \Sigma V^\top.$$
> The gradient of the spectral norm is given by:
> $$\frac{\partial||{\bf A}||_2}{\partial \bf A}=\textbf{u}_1\textbf{v}_1^\top, \quad \frac{\partial||{\bf A}||_2}{\partial {\bf A} _{ij}}=(\textbf{u}_1\textbf{v}_1^\top) _{ij},$$
> where $\textbf{u}_1$ is the left singular vector corresponding to the largest singular value, ${\bf v}_1$ is the right singular vector corresponding to the largest singular value.
>
> **In conclusion, SVD serves purely as a computational tool for gradient calculation for the first term in (11), it is an inherent component of (11).** We have updated the annotation for this part in the code, and will update more annotations to make this implementation detail clearer.
>
> ---
>
> **Q5(1): The concern becomes more serious with the following issue: While reviewing the code, another question arose: the confusion matrix is calculated on the full dataset instead of a batch of data, as shown in Lines 125–146 in train_fair.py. This seems inconsistent with existing methods (which calculate the confusion matrix within a batch)**
>
> **A5(1):**
> * **If "existing methods" include other robust fairness methods:** We would like to point out that FRL, WAT, and CFA all compute reweighting coefficient once per epoch (over training set or Validation set) rather than per minibatch over batch set.
> * **If "existing methods" just represent our method in (12):** Actually, (12) shows our method computes the confusion matrix over training set rather than batch set, i.e., $\mathcal{S}'$ in $\Psi(f_{\bf w},\mathcal{S}',\gamma)$.
>
> Specifically, we **compute the confusion matrix over training set** and **the computation of gradients involves two key terms**:
> 1. The gradient of the confusion matrix's spectral norm: $\frac{\partial || {\mathcal{C}}^{f _{\bf w}} _{{\mathcal{S}}',\gamma}|| _2}{\partial  ({\mathcal{C}}^{f _{\bf w}} _{{\mathcal{S}}',\gamma}) _{ij}}$. This is computed once per epoch over all training data using SVD, and the resulting gradient matrix is cached for use throughout the epoch.
> 2. The final KL gradient term in (11): $\frac{\partial (\mathcal{L}^{f _{\bf w}} _{\mathcal{S}',\gamma}) _{ij}}{\partial {\bf w}}$, which is computed over batch set due to the computational overload of processing the entire training set at once. We believe this operation is commonly used in DNNs' training.
>
> Then, model updates during each minibatch combine these two components.
>
> **In conclusion, our implementation aligns with the formulation in (12), where the confusion matrix is computed over the entire training set**. In practice, due to computational overload, we implement a hybrid approach to compute gradient: while the spectral norm gradient term in (11) is computed once per epoch using the full training set, the final KL gradient term is computed per minibatch due to computational constraints. **We are sorry for this misunderstanding and have provided more implementation details in App. F.**
>
> ---
>
> **Q5(2): and introduces extra training cost (about double the training time, as the adversarial examples are calculated twice).**

---

> ### Author Response · Authors · 2024-11-26
> **Response by Authors 2/3**
>
> **A5(2):** Our empirical results show that **the hybrid implementation achieves similar results to a full per-minibatch computation while being more computationally efficient due to reduced SVD calculations.** (They both compute adversarial examples twice.) We provide experimental results in the following table. We also compare our results with FAAL, as they reweight classes each minibatch. Note that in the table, **Hybrid** means the first spectral term in (11) is computed once per epoch over training set, and the final gradient term is comptued each minibatch over batch set. **Minibatch** means both terms are computed each minibatch over batch set.
>
> * Evaluation of different methods on CIFAR-10 with WRN-34-10. We report training time, fairness regularization on epoch/minibatch/Hybrid, and the average accuracy and the worst-class accuracy under standard AA, with baseline models of TRADES. The models are trained on 2 Nvidia A100 GPUs.
>
> | **Method**             |            | **AA (%)**   |            | **Worst (%)** |            | **Time/Epoch (s)** |            | **Regularize on**     |
> |-------------------------|------------|--------------|------------|----------------|------------|---------------------|------------|------------------------|
> | **TRADES**             |            | 52.51        |            | 23.20          |            | 664                 |            | N/A                    |
> | **+ FAAL$_{\text{AWP}}$** |          | 52.45        |            | 35.40          |            | 921                 |            | Minibatch              |
> | **+ Ours$_{\gamma=0.0}$** |          | **53.46**    |            | **36.30**      |            | 997             |            | Hybrid    |
> | **+ Ours$_{\gamma=0.0}$** |          | **53.39**    |            | **36.10**      |            | 1089            |            | Minibatch          |
>
> To address reviewer's potential concerns about **training time**, we have implemented and evaluated an alternative version of our method. Following CFA, we utilize adversarial examples from the previous epoch to generate the confusion matrix for the current epoch. The results are shown in the following table.
> | **Method**             |            | **AA (%)**   |            | **Worst (%)** |            | **Time/Epoch (s)** |            | **Regularize on**     |
> |-------------------------|------------|--------------|------------|----------------|------------|---------------------|------------|------------------------|
> | **+ Ours$_{\gamma=0.0}$** |          | **53.31**    |            | **36.10**      |            | 705             |            | Hybrid (Confusion matrix is generated by adversarial examples from previous epoch)    |
>
> As shown in the above table, this implementation achieves competitive results while increasing training time by only 6% compared to standard TRADES.
>
> ---
>
> **Q5(3): Calculating the confusion matrix from the full data decreases its bias with respect to the testing confusion matrix. It is now hard to distinguish the contributions between SVD and the method of confusion matrix calculation.**
>
> **A5(3):**
> * FRL, WAT, and CFA all compute reweighting coefficient once per epoch (over training set or Validation set) rather than per minibatch over batch set. As demonstrated in our manuscript, our method achieves superior results compared to these approaches.
> * We have discussed epoch/training set/minibatch/hybrid above, and compared ours with FAAL in minibatch level. Even when restricted to minibatch-level regularization, our method outperforms FAAL, demonstrating the effectiveness of our approach independent of computation scope.
> * SVD is used for computing the gradient of the spectral norm in (11) (It is an internal part of our method rather than an additional trick).
>
> We believe the concern about distinguishing contributions has been addressed.
>
> ---
>
> **Q6: Furthermore, my second question—'As a result, the proposed method may encounter issues such as sacrificing the performance of the best class and an increased likelihood of robust overfitting, both of which are evident in the empirical results presented in this paper'—is not well addressed. My primary concern is about the performance of the best class, rather than the average performance. Regarding the robust overfitting issue, it would be convincing if a training dynamics graph is provided.**
>
> **A6:**
>
> 1. **Our method and other robust fairness methods may or may not** reduce the best-class performance. As shown in the following table, ours may slightly increase or decrease the best-class performance. When maintaining average accuracy, improvements in worst-class performance typically require trade-offs with other classes, but not necessarily with the best-class. **While not central to our manuscript's claims, it's worth noting that our method often achieves better best-class performance compared to other robust fairness approaches.**

---

> ### Author Response · Authors · 2024-11-26
> **Response by Authors 3/3**
>
> | **Method**             |            | **AA (%)**   |            | **Worst (%)** |            | **Best (%)**       |
> |-------------------------|------------|--------------|------------|----------------|------------|--------------------|
> | **TRADES**             |            | 52.51        |            | 23.20          |            | 77.10             |
> | **+ FRL-RWRM$_{0.05}$**|            | 49.97        |            | 35.40          |            | 75.70             |
> | **+ FAAL$_{\text{AWP}}$** |          | 52.45        |            | 35.40          |            | 77.50             |
> | **+ Ours$_{\gamma=0.0}$** |          | 53.46    |            | 36.30      |            | 78.50         |
> |-------------------------|------------|--------------|------------|----------------|------------|--------------------|
> | **TRADES-AWP**         |            | 56.18        |            | 25.80          |            | 77.60             |
> | **+ FRL-RWRM$_{0.05}$**|            | 46.50        |            | 27.70          |            | 72.80             |
> | **+ FAAL$_{\text{AWP}}$** |          | 53.93        |            | 37.00          |            | 76.20             |
> | **+ Ours$_{\gamma=0.0}$** |          | 54.65    |            | 37.00      |            | 76.90         |
>
>
>
> 2. Due to time limit, we adversarially trained ResNet-18 models for standard TRADES and our method with $\gamma=0.0$ based on TRADES for 200 epochs using SGD with a momentum of 0.9, batch size of 256, weight decay of $5\times 10^{-4}$, and an initial learning rate of 0.1, which is reduced by a factor of 10 at the 100th and 150th epochs. Throughout the training process, we recorded both training and testing accuracy using PGD-10 attacks at each epoch. As illustrated in the **revised Fig. 4 in App. E.5**, both methods exhibit similar training and testing dynamics. These results suggest that our method does not independently introduce robust overfitting or address robust overfitting; rather, this phenomenon appears to be inherent to the fundamental adversarial training frameworks, e.g., TRADES, AWP, and AT.
>
> * **In conclusion**, our work focuses specifically on improving worst-class robust performance. While our method operates within adversarial training frameworks, we make no claims about causing or addressing robust overfitting, which is primarily determined by the underlying adversarial training methodology rather than fairness optimization approaches.
>
> ---
>
> Thank you once again for your further questions. If you have any additional questions or suggestions, we would be happy to address them.

---

> > ### Comment · Reviewer_RdH6 · 2024-11-27
> >
> > Thank you for providing the additional results. The explanations of SVD are clear, and the empirical findings on training costs are compelling. Regarding the best-class performance and the robust overfitting issue, while the training dynamics indicate that this phenomenon exists, I consider it a minor flaw that should not warrant rejecting this paper. Consequently, I have increased the soundness score and the final rating accordingly.

---

> > > ### Author Response · Authors · 2024-11-27
> > > **Thank the Reviewer RdH6**
> > >
> > > We are deeply grateful for the increased score and sincerely appreciate the significant time and effort you dedicated to reviewing our paper.
> > >
> > > We also greatly value your comments regarding robust overfitting and best-class performance. While our current work focuses on robust worst-class performance, we are excited about the prospect of developing a PAC-Bayesian framework that would jointly characterize worst-class, best-class, and robust generalization performance in future research.
> > >
> > > Thank you again for your valuable feedback.

---

### Official Review · Reviewer_ry7R · 2024-11-03

**Soundness:** 3
**Presentation:** 3
**Contribution:** 3
**Rating:** 8
**Confidence:** 3

**Summary:**

Recent studies have identified "robust fairness" as a significant challenge in deep learning, where the robust accuracy of DNNs can differ widely across various classes, affecting their reliability. Traditional methods attempt to tackle this issue by adjusting class weights during training, yet these methods often fail to predict which class will have the poorest robust accuracy in testing. To address this, researchers have developed a new theoretical framework within the PAC-Bayesian paradigm to better understand and control the worst-case robust error. This framework introduces a regularization technique aimed at reducing the spectral norm of the robust confusion matrix, thereby improving the robust accuracy of the most vulnerable class and promoting fairer outcomes across all classes.

**Strengths:**

1. The paper is well written and organized for readers' easy understanding.

2. The work is well motivated.

3. The performance is good.

**Weaknesses:**

1. The spectral norm should be denoted by another symbol rather than ||*||_2. It is rather confused between spectral norm and l_2 norm. Specifically, in Line 311 ~ Line 315, the authors mention the relationship of l_1 and l_2 norm, but the norm in Eqn. (8) should be spectral norm? So I not sure whether it is correct for the derivation.

**Questions:**

Please see Weakness.

---

> ### Author Response · Authors · 2024-11-22
> **Response by authors**
>
> Dear Reviewer ry7R,
>
> We sincerely thank you for your positive comments. In the following, we address your raised concern.
>
> ---
>
> **Q1: The spectral norm should be denoted by another symbol rather than $||\cdot||_2$. It is rather confused between spectral norm and l_2 norm. Specifically, in Line 311 ~ Line 315, the authors mention the relationship of l_1 and l_2 norm, but the norm in Eqn. (8) should be spectral norm? So I not sure whether it is correct for the derivation.**
>
> **A1:** We apologize for the confusion regarding notation and have revised our manuscript to consistently use "spectral norm of matrix" instead of "$\ell_2$ norm of matrix." Following standard convention in PAC-Bayes literature, we use capital letters (e.g., $\bf W$) for matrices and lowercase letters (e.g., $\bf w$) for vectors. Thus, $||{\bf W}||_2$ specifically denotes the spectral norm when applied to a matrix. While these notations are established in PAC-Bayes literature and cannot be easily modified, we have added clarifying explanations throughout the text to prevent misinterpretation.
>
> ---
>
> We appreciate your positive attitude towards this manuscript. Please let us know if you have any additional questions.
>
> Sincerely,
> Authors

---

> > ### Comment · Reviewer_ry7R · 2024-11-27
> > **response**
> >
> > Thank you for the authors’ responses, which have addressed my concerns. I have raised my score.

---

> > > ### Author Response · Authors · 2024-11-27
> > > **Thank the Reviewer ry7R**
> > >
> > > We are delighted to see our score has been improved to 8. Thank you for your time and dedication throughout this review process.

---

### Official Review · Reviewer_ZUyD · 2024-11-04

**Soundness:** 3
**Presentation:** 3
**Contribution:** 3
**Rating:** 6
**Confidence:** 1

**Summary:**

This paper endeavors to address the robust fairness problem and derives a robust generalization bound for the worst-class robust error within the PAC-bayesian framework, accounting for unknown data distributions.
Leveraging the insights gleaned from our theoretical results, they propose an effective and principled method to enhance robust fairness by introducing a spectral regularization term on the confusion matrix.

**Strengths:**

1. This work represents the first endeavor to develop a PAC-Bayesian framework to characterize the worst-class robust error across different classes. This is an important problem.
2. This paper is theoretically solid.
2. The improvement brought by this method is obvious in the experiment.

**Weaknesses:**

1. The fairness improvement method in this article seems to be designed only for adversarial training, so is there any improvement for the unfairness brought by other settings (such as long-tail training distribution)?
2. In the experiment, the author targeted AutoAttack. Does it have any effect on other attack methods?

**Questions:**

See weakness

---

> ### Author Response · Authors · 2024-11-22
> **Response by authors 1/2**
>
> Dear Reviewer ZUyD,
>
> We sincerely appreciate your comments about long-tail experiments and other attack types.
>
> In the following, we address the raised concerns one-by-one.
>
> ---
>
> **Q1: The fairness improvement method in this article seems to be designed only for adversarial training, so is there any improvement for the unfairness brought by other settings (such as long-tail training distribution)?**
>
> **A1:** We appreciate the reviewer's suggestion for broader empirical validation. Following Cui et al. (2024), we have extended our evaluation to include experiments on standard ImageNet and CIFAR-100 in clean training settings (detailed in Appendix E.2). For the reviewer's convenience, we present these new results here.
>
> | **Dataset**   |            | **Method**                 |            | **Easy (%)**      | **Medium (%)**    | **Hard (%)**        | **All (%)**      |
> |---------------|------------|----------------------------|------------|--------------------|--------------------|---------------------|------------------|
> | **ImageNet**  |            | Baseline (ResNet-50)      |            | 93.1              | 81.1              | 59.4               | 77.8            |
> |               |            | Menon et al. (2021)      |            | 91.7 (-1.4)       | 79.8 (-1.3)       | 61.4 (+2.0)        | 77.6            |
> |               |            | Cui et al. (2019)       |            | 91.6 (-1.5)       | 79.6 (-1.5)       | 61.3 (+1.9)        | 77.5            |
> |               |            | **Ours$_{\gamma=0.0}$**        |            | 91.5 (-1.6)   | 79.6 (-1.5)   | **62.3 (+2.9)**    | **77.8**        |
> | **CIFAR-100** |            | Baseline (WRN-34-10)      |            | 92.2              | 83.2              | 70.1               | 81.7            |
> |               |            | Menon et al. (2021)      |            | 91.5 (-0.7)       | 83.2 (+0.0)       | 70.4 (+0.3)        | 81.6            |
> |               |            | Nam et al. (2020)        |            | 90.4 (-1.8)       | 81.5 (-1.7)       | 67.7 (-2.4)        | 79.7            |
> |               |            | Liu et al., (2021)        |            | 91.3 (-0.9)       | 82.4 (-0.8)       | 69.6 (-0.5)        | 81.0            |
> |               |            | **Ours$_{\gamma=0.0}$**        |            | 91.3 (-0.9)   | 83.0 (-0.2)   | **71.2 (+1.1)**    | **81.8**        |
>
>
> The expanded experiments demonstrate that our method's effectiveness generalizes to larger-scale datasets: it successfully enhances fairness in clean training by improving test accuracy on "hard" subsets of both ImageNet and CIFAR-100 while maintaining overall accuracy. These results provide additional evidence for the broader applicability of our approach across different dataset scales and complexity levels.
>
> | **Method**          |            | **Many-Shot (%)** |            | **Medium-Shot (%)** |            | **Few-Shot (%)** |
> |----------------------|------------|-------------------|------------|---------------------|------------|------------------|
> | **Baseline**        |            | 66.1             |            | 38.4               |            | 8.9              |
> | **Ours$_{\gamma=0.0}$**  |            | 61.3         |            | 45.5           |            | 31.2         |
>
>
> In addition, we evaluate long-tail image classification on ImageNet-LT (Liu et al., 2019) using ResNeXt-50-32x4d as our baseline, trained with SGD (momentum 0.9, batch size 512) and cosine learning rate decay from 0.2 to 0.0 over 90 epochs.
> As shown in the table, our regularizer leads to improved fairness across the distribution: while showing a slight decrease in many-shot performance, it achieves significant gains in few-shot classes and moderate improvements in medium-shot classes, demonstrating effectiveness in addressing fairness problem in long-tail scenarios.
>
> [1] Jiequan Cui, Beier Zhu, Xin Wen, Xiaojuan Qi, Bei Yu, and Hanwang Zhang. Classes are not equal: An empirical study on image recognition fairness. In Proceedings of the IEEE/CVF Conference on Computer Vision and Pattern Recognition, pp. 23283–23292, 2024.
> [2] Aditya Krishna Menon, Sadeep Jayasumana, Ankit Singh Rawat, Himanshu Jain, Andreas Veit, and Sanjiv Kumar. Long-tail learning via logit adjustment. In International Conference on Learning Representations, 2021.
> [3] Yin Cui, Menglin Jia, Tsung-Yi Lin, Yang Song, and Serge Belongie. Class-balanced loss based on effective number of samples. In Proceedings of the IEEE/CVF conference on computer vision and pattern recognition, pp. 9268–9277, 2019.
> [4] Junhyun Nam, Hyuntak Cha, Sungsoo Ahn, Jaeho Lee, and Jinwoo Shin. Learning from failure: De-biasing classifier from biased classifier. Advances in Neural Information Processing Systems, 33:20673–20684, 2020.

---

> ### Author Response · Authors · 2024-11-22
> **Response by authors 2/2**
>
> [5] Evan Z Liu, Behzad Haghgoo, Annie S Chen, Aditi Raghunathan, Pang Wei Koh, Shiori Sagawa, Percy Liang, and Chelsea Finn. Just train twice: Improving group robustness without training group information. In International Conference on Machine Learning, pp. 6781–6792. PMLR, 2021.
> [6] Ziwei Liu, Zhongqi Miao, Xiaohang Zhan, Jiayun Wang, Boqing Gong, and Stella X Yu. Large scale long-tailed recognition in an open world. In Proceedings of the IEEE/CVF conference on computer vision and pattern recognition, pp. 2537–2546, 2019.
>
> ---
>
> **Q2: In the experiment, the author targeted AutoAttack. Does it have any effect on other attack methods?**
>
> **Q2:** We initially focused our evaluations on AutoAttack (AA) as it represents one of the strongest attack methods and is widely accepted as a comprehensive benchmark for adversarial robustness. However, we appreciate the reviewer's suggestion for more extensive evaluations.
>
> We provide more results in revised Tab. 4, with evaluation of different fine-tuning methods on CIFAR-10 with WRN-34-10. We report the average accuracy and the worst-class accuracy under standard $\ell_\infty$ norm PGD-20/CW-20 attack, with baseline models of TRADES and TRADES-AWP. For your convenience, we move the table here.
>
> | **Fine-Tuning Method**    |            | **PGD-20 (%)** | **Worst (%)** | **CW-20 (%)** | **Worst (%)** |            | **PGD-20 (%)** | **Worst (%)** | **CW-20 (%)** | **Worst (%)** |
> |---------------------------|------------|----------------|---------------|----------------|---------------|------------|----------------|---------------|----------------|---------------|
> |                           |            | **TRADES**     |               |                |               |            | **TRADES-AWP** |               |                |               |
> | **TRADES/AWP**            |            | 55.32          | 27.10         | 53.92          | 24.80         |            | **59.20**      | 28.80         | **57.14**      | 26.50         |
> | + FRL-RWRM$_{0.05}$       |            | 53.16          | 40.60         | 51.39          | 36.30         |            | 49.90          | 31.70         | 49.68          | 34.00         |
> | + FRL-RWRM$_{0.07}$       |            | 53.76          | 39.20         | 52.92          | 36.80         |            | 48.63          | 30.90         | 49.77          | 31.50         |
> | + FAAL$_{\text{AT}}$      |            | 53.46          | 39.80         | 52.72          | 38.20         |            | 52.54          | 35.00         | 51.70          | 34.40         |
> | + FAAL$_{\text{AWP}}$     |            | 56.07          | 43.30         | 54.16          | 38.60         |            | 57.14          | 43.40         | 55.34          | 40.10         |
> | **+ Ours$_{\gamma=0.0}$** |            | **57.83**      | **44.60**     | **56.09**      | **39.70**     |            | 59.06          | 44.10         | 56.79          | 41.30         |
> | **+ Ours$_{\gamma=0.1}$** |            | 57.66          | 44.10         | 55.97          | 38.90         |            | 58.87          | **44.50**     | 56.44          | **41.80**     |
>
> As shown in the table, our method maintains superior worst-class performance across these different attack types, and gets better average robust performance compared with other robust fairness methods.
>
> ---
>
> We appreciate your useful suggestions and hope we have addressed your concerns. Please let us know if you have any additional questions.
>
> Sincerely,
> Authors

---

> > ### Comment · Reviewer_ZUyD · 2024-11-25
> >
> > Thanks to the author for the response, all my concerns are solved.
> >
> > Since I'm not an expert in this field, I decided to keep my score and confidence.

---

> > > ### Author Response · Authors · 2024-11-26
> > > **Thank you**
> > >
> > > We sincerely appreciate your review and are glad that we have successfully addressed your concerns. Your suggestions have really enhanced our manuscript's quality. Thank you for your time and dedication throughout this review process.

---

### Official Review · Reviewer_dWn3 · 2024-11-04

**Soundness:** 3
**Presentation:** 3
**Contribution:** 4
**Rating:** 6
**Confidence:** 4

**Summary:**

This work derives a robust generalization bound for the worst-case (i.e. adversarial) robust error using a PAC-Bayesian analysis. This is the worst-margin violation in an epsilon-perturbed adversarial input setup, which can be represented as L1 norm on a perturbed confusion matrix. Through a link to the L2 norm (with an unknown empirically tight ratio), the bound further splits to (1) spectral norm of the empirical margined confusion matrix and (2) a component that depends on the information content of the input data and the specifics of the hypothesis space.

The paper focuses on the spectral norm component as much of the recent literature has focused on the other component. The work argues that this spectral norm can be used as a regularization term during training, and provides an approximation of it with a feasible implementation path.

The proposed method is evaluated on the worst-case accuracy (test) metric under both clean-data and AutoAttack (Groce & Hein, 2020b) settings, and is compared to a number of common adversarial training methods. Experiments on image datasets show that the proposed method has comparable results to SOTA on average metrics while outperforming most methods on worst-case robust error. Transfer learning experiments (CIFAR to ImageNet) show that the regularized fine-tuning outperforms the baseline on the worst-case robust error metric.

**Strengths:**

- The approximation in section 4.1 makes it possible to implement an adversarial spectral-norm regularizer effectively.

- Based on limited empirical analysis, the method looks robust to the choice of hyperparameters.

- The method performs well against SOTA attack methods (AutoAttack), and not just the limited setting considered in the theoretical analysis.

**Weaknesses:**

- Empirical analysis is somewhat limited both in domain and the datasets used. It’s hard to argue only based on the provided analysis that the results will extend to larger models, other vision datasets, or non-vision classification tasks. Adding experiments with much larger datasets (e.g. larger ImageNet) or non-vision tasks would greatly improve the empirical analysis in the paper.

**Questions:**

- The assumption on the norm of U (i.e. restricting f_{w+u}(x) - f_w(x), instead of bounding the input perturbation) seems crucial to the derivation of Lemma 3.2. Have you looked at the value of this norm in your experiments? An analysis of the norm of U for some of the experiments conducted for the work could help establish how tightly this assumption is satisfied in practice.

---

> ### Author Response · Authors · 2024-11-22
> **Response by authors 1/2**
>
> Dear Reviewer dWn3,
>
> We sincerely appreciate your comments about experiments, typo, and the assumption of $\bf u$.
>
> We have revised the typo in the updated manuscript, in the following, we address other concerns one-by-one.
>
> ---
>
> **Q1: Empirical analysis is somewhat limited.**
>
> **A1:** We appreciate the reviewer's suggestion for broader empirical validation. Following Cui et al. (2024), we have extended our evaluation to include experiments on standard ImageNet and CIFAR-100 in clean training settings (detailed in Appendix E.2). For the reviewer's convenience, we present these new results here.
>
> | **Dataset**   |            | **Method**                 |            | **Easy (%)**      | **Medium (%)**    | **Hard (%)**        | **All (%)**      |
> |---------------|------------|----------------------------|------------|--------------------|--------------------|---------------------|------------------|
> | **ImageNet**  |            | Baseline (ResNet-50)      |            | 93.1              | 81.1              | 59.4               | 77.8            |
> |               |            | Menon et al. (2021)      |            | 91.7 (-1.4)       | 79.8 (-1.3)       | 61.4 (+2.0)        | 77.6            |
> |               |            | Cui et al. (2019)       |            | 91.6 (-1.5)       | 79.6 (-1.5)       | 61.3 (+1.9)        | 77.5            |
> |               |            | **Ours$_{\gamma=0.0}$**        |            | 91.5 (-1.6)   | 79.6 (-1.5)   | **62.3 (+2.9)**    | **77.8**        |
> | **CIFAR-100** |            | Baseline (WRN-34-10)      |            | 92.2              | 83.2              | 70.1               | 81.7            |
> |               |            | Menon et al. (2021)      |            | 91.5 (-0.7)       | 83.2 (+0.0)       | 70.4 (+0.3)        | 81.6            |
> |               |            | Nam et al. (2020)        |            | 90.4 (-1.8)       | 81.5 (-1.7)       | 67.7 (-2.4)        | 79.7            |
> |               |            | Liu et al., (2021)        |            | 91.3 (-0.9)       | 82.4 (-0.8)       | 69.6 (-0.5)        | 81.0            |
> |               |            | **Ours$_{\gamma=0.0}$**        |            | 91.3 (-0.9)   | 83.0 (-0.2)   | **71.2 (+1.1)**    | **81.8**        |
>
>
> The expanded experiments demonstrate that our method's effectiveness generalizes to larger-scale datasets: it successfully enhances fairness in clean training by improving test accuracy on "hard" subsets of both ImageNet and CIFAR-100 while maintaining overall accuracy. These results provide additional evidence for the broader applicability of our approach across different dataset scales and complexity levels.
>
> | **Method**          |            | **Many-Shot (%)** |            | **Medium-Shot (%)** |            | **Few-Shot (%)** |
> |----------------------|------------|-------------------|------------|---------------------|------------|------------------|
> | **Baseline**        |            | 66.1             |            | 38.4               |            | 8.9              |
> | **Ours$_{\gamma=0.0}$**  |            | 61.3         |            | 45.5           |            | 31.2         |
>
>
> In addition, we evaluate long-tail image classification on ImageNet-LT (Liu et al., 2019) using ResNeXt-50-32x4d as our baseline, trained with SGD (momentum 0.9, batch size 512) and cosine learning rate decay from 0.2 to 0.0 over 90 epochs.
> As shown in the table, our regularizer leads to improved fairness across the distribution: while showing a slight decrease in many-shot performance, it achieves significant gains in few-shot classes and moderate improvements in medium-shot classes, demonstrating effectiveness in addressing fairness problem in long-tail scenarios.
>
> [1] Jiequan Cui, Beier Zhu, Xin Wen, Xiaojuan Qi, Bei Yu, and Hanwang Zhang. Classes are not equal: An empirical study on image recognition fairness. In Proceedings of the IEEE/CVF Conference on Computer Vision and Pattern Recognition, pp. 23283–23292, 2024.
> [2] Aditya Krishna Menon, Sadeep Jayasumana, Ankit Singh Rawat, Himanshu Jain, Andreas Veit, and Sanjiv Kumar. Long-tail learning via logit adjustment. In International Conference on Learning Representations, 2021.
> [3] Yin Cui, Menglin Jia, Tsung-Yi Lin, Yang Song, and Serge Belongie. Class-balanced loss based on effective number of samples. In Proceedings of the IEEE/CVF conference on computer vision and pattern recognition, pp. 9268–9277, 2019.
> [4] Junhyun Nam, Hyuntak Cha, Sungsoo Ahn, Jaeho Lee, and Jinwoo Shin. Learning from failure: De-biasing classifier from biased classifier. Advances in Neural Information Processing Systems, 33:20673–20684, 2020.
> [5] Evan Z Liu, Behzad Haghgoo, Annie S Chen, Aditi Raghunathan, Pang Wei Koh, Shiori Sagawa, Percy Liang, and Chelsea Finn. Just train twice: Improving group robustness without training group information. In International Conference on Machine Learning, pp. 6781–6792. PMLR, 2021.

---

> ### Author Response · Authors · 2024-11-22
> **Response by authors 2/2**
>
> [6] Ziwei Liu, Zhongqi Miao, Xiaohang Zhan, Jiayun Wang, Boqing Gong, and Stella X Yu. Large scale long-tailed recognition in an open world. In Proceedings of the IEEE/CVF conference on computer vision and pattern recognition, pp. 2537–2546, 2019.
>
> ---
>
> **Q2: The assumption on the norm of U (i.e. restricting f_{w+u}(x) - f_w(x), instead of bounding the input perturbation) seems crucial to the derivation of Lemma 3.2. Have you looked at the value of this norm in your experiments? An analysis of the norm of U for some of the experiments conducted for the work could help establish how tightly this assumption is satisfied in practice.**
>
> **A2:** Our approach follows established PAC-Bayesian frameworks (Neyshabur et al. (2017); Farnia et al. (2019)), where the norm restriction on $\bf u$ serves as a theoretical bridge connecting worst-class error with the spectral norm of empirical confusion matrix and model weights.
>
> The perturbation degree of $\bf u$ reflects the sharpness/flatness of the base classifier $f_{\bf w}$. Its effectiveness is usually evaluated in relation to generalization gaps (Jiang et al. (2020)), as the single value of $\bf u$ is un-imformative. While Sec. 3 presents weight-norm-based bounds, we further validate the PAC-Bayesian bound/assumption empirically using Jiang et al. (2020)'s sharpness-based sampling method:
>
> 1. Sample 50 perturbations ${\bf u}_1, ..., {\bf u}_{50}$ from $\mathcal{N}(0,\sigma^2 \bf I)$
> 2. Find the largest $\sigma^2 \in \{0.01, 0.02, ..., 1\}$ where training accuracy drop between $f_{\bf w}$ and any $f_{{\bf w}+{\bf u_i}}$ stays within 5%
>
> We have provided the results in revised **App. E.3**, for your convenience, we also put the table here. The models are trained from scratch on CIFAR-10 using Preact-ResNet18 model with $\ell_\infty$ norm attack.
>
> | **Method**          |            | **Training AA - Test AA (%)** |            | **Test Worst-Class AA (%)** |            | **Largest Variance** |
> |----------------------|------------|-------------------------------|------------|-----------------------------|------------|-----------------------|
> | **PGD-AT**          |            | 7.53                         |            | 12.90                       |            | 0.14                  |
> | **TRADES**          |            | 6.08                         |            | 20.70                       |            | 0.17                  |
> | **Ours$_{\gamma=0.0}$**  |            | 5.47                     |            | 33.80                   |            | 0.21              |
>
>
> Our results show that smaller generalization gaps and higher worst-class accuracy correspond to larger variances, indicating smoother base classifiers. These findings align with previous work (Jiang et al. (2020), Xiao et al. (2023)), supporting the effectiveness of the PAC-Bayesian framework.
>
> [1] Behnam Neyshabur, Srinadh Bhojanapalli, and Nathan Srebro. A pac-bayesian approach to spectrally-normalized margin bounds for neural networks. arXiv preprint arXiv:1707.09564, 2017.
> [2] Farzan Farnia, Jesse M Zhang, and David Tse. Generalizable adversarial training via spectral normalization. In ICLR, 2019.
> [3] Yiding Jiang, Behnam Neyshabur, Hossein Mobahi, Dilip Krishnan, and Samy Bengio. Fantastic generalization measures and where to find them. International Conference on Learning Representations, 2020.
> [4] Jiancong Xiao, Ruoyu Sun, and Zhi-Quan Luo. Pac-bayesian spectrally-normalized bounds for adversarially robust generalization. Advances in Neural Information Processing Systems, 2023.
>
> ---
>
> We appreciate your insightful comments and hope we have addressed your concerns. Please let us know if you have any additional questions.
>
> Sincerely,
> Authors

---

> > ### Comment · Reviewer_dWn3 · 2024-11-25
> > **Review on Updates**
> >
> > The authors have expanded the experimental analysis to include a larger/harder problem domain, on which the proposed method compared favorably against SOTA. Another experiment is added to study the sharpness/flatness of the base classifier as it relates to the generalization gap. I suggest the authors discuss the relationship between the perturbation degree of $u$ and the sharpness of base classifier $f$ in the main body of the paper.
> >
> > The added experiments and discussions address several of the issues I raised in my original review. I have thus updated my assessment to reflect these changes.

---

> ### Author Response · Authors · 2024-11-26
> **Thank the Reviewer dWn3**
>
> We sincerely appreciate your thorough review and are glad that we have successfully addressed your concerns, leading to an improved contribution score of 4. Your insightful suggestions have substantially enhanced our manuscript's quality. Following your feedback, we will move discussions about sharpness into the main body of the manuscript. Thank you for your time and dedication throughout this review process.

---

### Official Review · Reviewer_rown · 2024-11-04

**Soundness:** 3
**Presentation:** 4
**Contribution:** 3
**Rating:** 8
**Confidence:** 3

**Summary:**

The paper studies the robust fairness of deep neural networks. The authors first derive a robust generalization bound for the worst-class robust error within the PAC-Bayesian framework. The bound comprises two key components: the spectral norm of the empirical robust confusion matrix and a model-dependent and training data-dependent term. Based on this, the authors further develop a novel regularization technique to optimize the spectral norm of the robust confusion matrix to improve the robust fairness. Extensive experiments are conducted to show the effectiveness of the method.

**Strengths:**

- The paper is well-written and easy to follow. It gives adequate background on the topics of fairness, adversarial robustness, and PAC-bound, which can be very helpful for readers who are not familiar with the field to understand the gist of the paper.
- The contribution is quite significant because:
1. this is the first PAC-Bayesian framework to characterize the worst-class robust error across different classes.
2. the proposed regularization term is novel because it aims to improve the robust fairness from the spectral norm of the robust confusion matrix, which is ignored by previous methods.
3. during evaluation, the method shows improved results in terms of robust fairness across all settings.

**Weaknesses:**

- The motivation from the intro feels disconnected from the method. In Figure 2, the authors argue that the class exhibiting the worst robust performance on the training set may not be the same as the one on the test set. However, how would the proposed method in Section 4 address this issue? It would be great if the authors could provide some discussion about the connection between this motivation and the proposed method.

- It would be great to also see some analysis for the hyper-parameters $\alpha$ and $\gamma$ for a better understanding of the method. How optimal is 0.3 for $\alpha$? What is the effect of changing it? Does it trade off the average accuracy vs the worst class robustness? I would appreciate if the authors can provide some intuition behind the effect of these two hyper-parameters and also some experimental results showing how AA & Worst changes with respect of different values.

- Typos:
1. In Equation 10, (x, y) -> (x', y)
2. Line 425, W -> We

**Questions:**

- This question relates to the motivation in Fig. 2. What is the correlation between the class-wise robust accuracy of the training and the test set? I understand the robust confusion matrix over the training set does not always align with that of the test set. But if they are aligned most of the time, should it still be an important concern?

- It seems that $\gamma=0$ always has slightly better AA and worse Worst compared to $\gamma=0.1$. Why is this the case?

- The derivation from Eq. 9 to Eq. 11 is a bit confusing. Why is the middle term approximated by the sign? How good/bad is this approximation? It would be great if the authors could provide a bit detailed explanation for lines 354-358.

- Why is the best result in the 5th column of Tab. 1 the original TRADES-AWP? It seems that all the fine-tuning methods fail to improve the average accuracy?

---

> ### Author Response · Authors · 2024-11-22
> **Response by authors 1/3**
>
> Dear Reviewer rown,
>
> We sincerely appreciate your comments about motivation, hyper-parameter, and the derivation of (11).
>
> We have revised the typos in the updated manuscript, in the following, we address the raised concerns one-by-one.
>
> ---
>
> **Q1(1): The motivation from the intro feels disconnected from the method.; Fig. 2.**
>
> **A1(1):** We apologize for any confusion and have revised the third paragraph to better articulate the connection between our motivation and method. Our key observation is that existing robust fairness approaches (e.g., FRL and FAAL) explicitly reweight classes based on training set performance during adversarial training. However, we find that different classes exhibit varying generalization gaps between training and test set performance. This class-dependent divergence means that training set performance may not reliably predict test set performance, potentially limiting the effectiveness of previous explicit reweighting methods. Moreover, as demonstrated in revised **Fig. 2 (right)**, these methods can actually amplify the training-test divergence **(lower training-test correlation)**, consequently hindering their ability to optimize worst-class robust accuracy.
>
> Our approach takes a fundamentally different path: instead of explicitly reweighting classes based on empirical class performance, we regularize the spectral norm of the empirical confusion matrix to improve worst-class robust performance. This approach is theoretically grounded in PAC-Bayesian bounds and, empirically, achieves both lower training-test divergence (higher training-test correlation, see Fig. 2) and better worst-class robust performance compared to FRL and FAAL.
>
> In conclusion, while training-test divergence can limit the effectiveness of explicit class-reweighting methods (and may be exacerbated by them), our theoretically-motivated spectral norm regularization provides an alternative that avoids these pitfalls. Our empirical results demonstrate that this approach not only maintains better alignment between training and test performance but also achieves superior worst-class robust accuracy. **Note that our method is not primarily designed to address the training-test divergence, which may be inherent to the dataset, but rather to avoid explicit reweighting that can exacerbate this divergence and limit effectiveness.**
>
> **Justification:** (1) The empirical results in revised Fig. 2;
> (2) The explicit reweighting strategy proposed in FRL, FAAL, ...
> (3) The spectral regularizer (distinguish from explicit reweighting) shown in this manuscript
> (4) The experiments shown in this manuscript
>
> **Q1(2): What is the correlation between the class-wise robust accuracy of the training and the test set?**
>
> **A1(2):** We analyze the class-wise robust accuracy correlation between training and test sets across four models: normal adversarially trained model, FRL fine-tuned model, FAAL fine-tuned model, and our method. As shown in revised **Fig. 2 (right)**, we quantify this relationship using both training-test covariance and Kendall rank correlation. The results demonstrate that while the normal model and our method maintain higher training-test correlation (lower divergence), models fine-tuned with FAAL and FRL exhibit significantly lower correlation, indicating they amplify the training-test divergence. **Thus, the main concern is that these reweighting methods can amplify this divergence and limit their effectiveness to improve worst-class performance.**
>
> **Q1(3): I understand the robust confusion matrix over the training set does not always align with that of the test set. But if they are aligned most of the time, should it still be an important concern?**
>
> **A1(3):** While normal adversarial training indeed maintains relatively low training-test divergence, our concern focuses on the behavior of explicit reweighting methods (e.g., FRL and FAAL). These methods, despite their intended purpose, can actually amplify the training-test divergence (see Fig. 2) and consequently limit their effectiveness in improving worst-class performance. Our method takes a different approach by avoiding direct class reweighting based on class-wise performance. As a result, it not only maintains the naturally low training-test divergence but also achieves better worst-class performance.
>
> ---
>
> **Q2(1): Analysis for the hyper-parameters; It seems that $\gamma=0.0$ always has slightly better AA and worse Worst compared to $\gamma=0.1$. Why is this the case?**
>
> **A2(1):** We have conducted a comprehensive analysis of different $\alpha,\gamma$ values ranging from 0.0 to 0.4, with complete results presented in revised **App. E.1**. For your convenience, we move the results here.

---

> ### Author Response · Authors · 2024-11-22
> **Response by authors 2/3**
>
> | **$\gamma$ = 0.0**     |            | **AA (%)** | **Worst (%)** |        | **$\alpha$ = 0.3**    |            | **AA (%)** | **Worst (%)** |
> |------------------|------------|------------|---------------|--------|----------------|------------|------------|---------------|
> | **$\alpha$**           |            |            |               |        | **$\gamma$**          |            |            |               |
> | 0.0             |            | 47.38      | 12.90         |        | 0.0            |            | **49.92**  | 33.80         |
> | 0.1             |            | 49.02      | 21.50         |        | 0.1            |            | 49.73      | **34.90**     |
> | 0.2             |            | 49.76      | 24.40         |        | 0.2            |            | 49.31      | 34.70         |
> | 0.3             |            | **49.92**  | **33.80**     |        | 0.3            |            | 48.56      | 33.90         |
> | 0.4             |            | 49.13      | 30.10         |        | 0.4            |            | 47.68      | 33.20         |
>
> * We investigate how regularization hyper-parameters $\alpha$ and $\gamma$ influence model performance through experiments on CIFAR-10 with PreAct ResNet-18. Models are trained for 200 epochs with batch size 128, using two configurations: (1) fixed $\gamma=0.0$ with $\alpha$ varying from 0.0 to 0.4, and (2) fixed $\alpha=0.3$ with $\gamma$ varying from 0.0 to 0.4.
> * As shown in the table (left), both average and worst-class AA accuracy initially increase with $\alpha$ before declining due to the disruption of training performance at larger values, leading to our choice of $\alpha=0.3$ for subsequent experiments.
>
> **Q2(2): It seems that $\gamma=0.0$ always has slightly better AA and worse Worst compared to $\gamma=0.1$. Why is this the case?**
>
> **A2(2):**
> * The table (right) demonstrates that with increasing $\gamma$, average AA accuracy consistently decreases while worst-class AA accuracy shows an initial improvement followed by deterioration. This behavior can be explained by two competing effects: while larger $\gamma$ values may impair overall training performance, they also reduce the influence of the final term in our bound, allowing the regularization term to more effectively optimize worst-class AA accuracy. This trade-off explains why $\gamma=0.1$ achieves better worst-class performance while maintaining acceptable average performance.
>
> ---
>
> **Q3: The derivation from Eq. 9 to Eq. 11 is a bit confusing.**
>
> **A3:** We appreciate the reviewer's comment and have revised our presentation in Sec. 4.1. Specifically, we have rewritten (9) and (11) to use $\sum$ notation for unit gradient summation rather than the potentially misleading $\frac{vector}{vector}$ notation.
>
> Revised (9):
> $$
> \underbrace{\frac{\partial || {\mathcal{C}}^{f _{\bf w}} _{{\mathcal{S}}',\gamma}|| _2}{\partial {\bf w}}} _{\textbf{Expensive}} \Longrightarrow
> \sum _{i\ne j}\underbrace{\frac{\partial || {\mathcal{C}}^{f _{\bf w}} _{{\mathcal{S}}',\gamma}|| _2}{\partial  ({\mathcal{C}}^{f _{\bf w}} _{{\mathcal{S}}',\gamma}) _{ij}}} _{\textbf{Cheap}} \times \underbrace{\frac{\partial ({\mathcal{C}}^{f _{\bf w}} _{{\mathcal{S}}',\gamma}) _{ij}}{\partial{\bf w}}} _{\textbf{Expensive}} \Longrightarrow
> \sum _{i\ne j}\underbrace{\frac{\partial || {\mathcal{C}}^{f _{\bf w}} _{{\mathcal{S}}',\gamma}|| _2}{\partial ({\mathcal{C}}^{f _{\bf w}} _{{\mathcal{S}}',\gamma}) _{ij}}} _{\textbf{Cheap}} \times \underbrace{\frac{\partial ({\mathcal{C}}^{f _{\bf w}} _{{\mathcal{S}}',\gamma}) _{ij}}{\partial ({\mathcal{L}}^{f _{\bf w}} _{{\mathcal{S}}',\gamma}) _{ij}}} _{\textbf{Approximate}} \times \underbrace{\frac{\partial ({\mathcal{L}}^{f _{\bf w}} _{{\mathcal{S}}',\gamma}) _{ij}}{\partial {\bf w}}} _{\textbf{Cheap}}
> $$
>
>
> Revised (11):
> $$
> \frac{\partial \Psi(f _{\bf w},\mathcal{S}',\gamma)}{\partial {\bf w}} = \sum _{i\ne j} \Bigg [ \frac{\partial || \mathcal{C}^{f _{\bf w}} _{\mathcal{S}',\gamma}|| _2}{\partial (\mathcal{C}^{f _{\bf w}} _{\mathcal{S}',\gamma}) _{ij}} \times \text{sign}\Bigg( \frac{\partial (\mathcal{C}^{f _{\bf w}} _{\mathcal{S}',\gamma}) _{ij}}{\partial (\mathcal{L}^{f _{\bf w}} _{\mathcal{S}',\gamma}) _{ij}}\Bigg) \times \frac{\partial (\mathcal{L}^{f _{\bf w}} _{\mathcal{S}',\gamma}) _{ij}}{\partial {\bf w}} \Bigg ]
> $$

---

> ### Author Response · Authors · 2024-11-22
> **Response by authors 3/3**
>
> In the right-hand side of (9), we approximate the general direction (sign) of $\partial (\mathcal{C}^{f _{\bf w}} _{\mathcal{S}',\gamma}) _{ij}/\partial ({\mathcal{L}}^{f _{\bf w}} _{\mathcal{S}',\gamma}) _{ij}$ based on the observation that the descent direction of non-diagonal elements in $\mathcal{C}^{f _{\bf w}} _{\mathcal{S}',\gamma}$ closely aligns with that of ${\mathcal{L}}^{f{\bf w}} _{\mathcal{S}',\gamma}$. This leads to our approximation: $$\text{sign}\Bigg( \frac{\partial (\mathcal{C}^{f _{\bf w}} _{\mathcal{S}',\gamma}) _{ij}}{\partial (\mathcal{L}^{f _{\bf w}} _{\mathcal{S}',\gamma}) _{ij}}\Bigg)\approx 1.$$
> This approximation strategy parallels common practice in machine learning. For example, in classification tasks, we cannot directly compute gradients with respect to error (accuracy), so we optimize differentiable proxies like cross-entropy loss or KL divergence, assuming their optimization directions align. Similarly, we adopt this approximation for computational efficiency, as accurately computing the exact gradient term would be prohibitively complex.
>
> **In conclusion**, the gradient inside the sign function cannot be used directly due to the non-differentiable nature of 0-1 errors in $\mathcal{C}^{f _{\bf w}} _{\mathcal{S}',\gamma}$ (high complexity to estimate its discrete gradient). Therefore, we approximate the gradient direction using the sign() function, effectively treating it as 1. We would like to point out that this approximation strategy is commonly used in machine learning, e.g., optimize differentiable loss functions (e.g., cross-entropy, KL divergence) to reduce non-differentiable error.
>
> ---
>
> **Q4: Why is the best result in the 5th column of Tab. 1 the original TRADES-AWP? It seems that all the fine-tuning methods fail to improve the average accuracy?**
>
> **A4:** The observation is correct - AWP, being one of the strongest adversarial training methods, achieves the best average AA accuracy. While fine-tuning methods may not improve upon this baseline average performance, our work primarily focuses on enhancing worst-class performance. Importantly, when compared with other fairness methods applied to AWP-trained models, our approach achieves both the best worst-class AA accuracy and the highest average AA accuracy among all fine-tuning methods.
>
> ---
>
> We appreciate your thoughtful feedback and hope we have addressed your concerns. Please let us know if you have any additional questions.
>
> Sincerely,
> Authors

---

> > ### Comment · Reviewer_rown · 2024-12-01
> >
> > I thank the authors for the detailed responses, which address my concerns. Considering the significance of the work and the strengths mentioned by other reviewers, I increased my score.

---

> > > ### Author Response · Authors · 2024-12-01
> > > **Thank the Reviewer rown**
> > >
> > > We are glad to see our score has been improved to 8. Your thoughtful suggestions have significantly enhanced the quality of our manuscript. Thank you for your time and dedication throughout this review process.

---

### Author Response · Authors · 2024-11-23
**General Response by Authors**

We sincerely thank the reviewers for their thorough feedback and insightful comments.

**We have carefully addressed each point raised and incorporated all constructive suggestions into the revised manuscript, with modifications highlighted in blue text.**

Should any aspects require further clarification, we welcome additional questions. If the reviewers find that their concerns have been satisfactorily addressed, we would appreciate their consideration in updating the evaluation score to reflect these improvements.

---

### Comment · Area_Chair_hpyr · 2024-11-25
**Interactive Discussions**

Dear Reviewers,

Thank you for your efforts in reviewing this paper. We highly encourage you to participate in interactive discussions with the authors before November 26, fostering a more dynamic exchange of ideas rather than a one-sided rebuttal.

Please feel free to share your thoughts and engage with the authors at your earliest convenience.

Thank you for your collaboration.

Best regards, ICLR 2025 Area Chair

---

### Meta-Review · Area_Chair_hpyr · 2024-12-20

**Metareview:**

This submission tackles the issue of fairness in adversarial robustness by leveraging PAC-based bounds to analyze worst-class error. Specifically, it first derives a bound for deterministic classifiers within the PAC framework. It then applies techniques from Neyshabur et al. to bound the KL divergence, which is typically challenging to compute. Finally, the analysis from Xiao et al. is used to obtain Lemma 3.4. Empirically, based on their analysis of the confusion matrix, the authors propose the addition of an approximate regularization term, and experimental results demonstrate the effectiveness of their approach.

After the rebuttal, all reviews recommended acceptance. The area chair concurs, noting that this submission makes a solid contribution to the fields of fairness and adversarial robustness, and recommends acceptance.

**Additional Comments On Reviewer Discussion:**

Most concerns were addressed during the rebuttal phase, and the authors are encouraged to incorporate these discussions into the final version.

---

### Decision · Program_Chairs · 2025-01-22

Accept (Poster)